# CAN LARGE LANGUAGE MODELS EFFECTIVELY MODIFY GRAPHS?

## ABSTRACT

Graphs are essential tools for modeling complex relationships. While prior research with earlier generations of large language models (LLMs) showed them to struggle with basic graph primitives, we find that the situation has changed with modern state-of-the-art (SOTA) LLMs, which excel at these tasks. Given these advances, we propose a more challenging evaluation problem: graph modification, a foundational, interpretable, and non-trivial problem in which an LLM must determine the outcome of adding or deleting a given sequence of nodes or edges, and potentially then compute on the resulting modified graph. We introduce GraphModQA, a novel benchmark dataset comprising graph modification question-answer pairs designed to rigorously test LLMs' abilities in graph manipulation and dynamic reasoning. Our results show that while SOTA LLMs perform well on static graph property tasks, their accuracy degrades on graph modification tasks; their performance is particularly low as the number of modifications increases, and when the adjacency matrix is used to represent the graph — an essential encoding not explored in previous work. We provide new techniques for improving performance on graph modification tasks, and we introduce Modify-and-Print (MAP) prompting, which asks models to output the intermediate adjacency matrices at each step, and which markedly improves the models' performance. Our findings highlight a critical gap in current LLM capabilities regarding dynamic graph reasoning tasks and underscore the potential of techniques like MAP prompting to mitigate these challenges.

## 1 INTRODUCTION

Large Language Models (LLMs) have revolutionized various aspects of natural language processing, demonstrating remarkable capabilities in understanding and generating human-like text (Brown et al. (2020)). Despite their success, the intersection of LLMs and graph-based tasks remains understudied. Graphs are fundamental for modeling complex relationships in domains such as social networks and recommender systems (Schneider et al. (2022); Wu et al. (2022)). Understanding how LLMs can be leveraged to process and reason about graph structures is crucial for advancing their applicability in these areas.

Previous work by Wang et al. (2024a) explored the ability of pretrained LLMs to handle a variety of graph tasks. This study included tasks such as calculating shortest paths and simulating graph neural networks, highlighting the challenges LLMs face with graph-structured data. Building upon this, Fatemi et al. (2023) introduced GraphQA, a synthetic dataset of graph property question-answer pairs, which included additional and more fundamental graph property tasks, such as calculating the number of nodes in the graph or the degree of a particular node, noting the simplicity and interpretability of these tasks compared to those in Wang et al. (2024a). Through these experiments, Fatemi et al. (2023) concluded that LLMs, in particular models from the PaLM family (Anil et al. (2023)) performed poorly on fundamental graph property tasks when provided with various graph encoding functions such as incident lists or textual descriptions, indicating a limitation in their ability to process and reason about structured data represented in graphs.

While prior research with earlier generations of large language models (LLMs) showed them to struggle with basic graph primitives, it is unknown how modern state-of-the-art (SOTA) LLMs perform on these property tasks. Modern LLMs have shown huge performance increases across many

reasoning tasks compared to PaLM (Dubey et al. (2024)), suggesting the possibility that these performance increases may translate into the domain of graph reasoning. Our empirical analysis reveals that contemporary state-of-the-art (SOTA) LLMs now excel at basic graph property tasks. Models such as o1-mini and Llama 3.1 405B demonstrate proficiency in identifying patterns and making simple inferences from graph data across different encoding methods.

Given these advances, we propose a more challenging evaluation problem: graph modification. Unlike static graph property tasks, graph modification requires models to perform a sequence of operations—such as adding or removing nodes or edges—and then answer questions about the resulting graph or output the modified graph itself. These tasks, which have yet to be studied in the context of LLMs, are foundational and interpretable yet non-trivial, as they necessitate maintaining and updating an internal representation of the graph through each modification step. The task of outputting the final modified graph is especially complex due to the intricacies of the output space. In real-world applications, graphs are rarely static; they evolve over time with the addition or removal of nodes and edges. This dynamic nature is evident in domains like social network analysis, where relationships and interactions constantly change (Kazemi et al. (2020)), and in evolving knowledge bases that need to adapt to new information (Trivedi et al. (2017); Schneider et al. (2022)). To fully assess the graph reasoning capabilities of modern LLMs, it is essential to evaluate their ability to understand and manipulate graphs that undergo modifications. Reasoning on these graphs combines the inherent difficulty of dynamic state maintenance with high-level reasoning about the final modified graph, making it significantly more rigorous in evaluating a model's capability to handle evolving graph structures.

We introduce **GraphModQA**, a novel benchmark dataset containing graph modification question-answer pairs. GraphModQA is designed to rigorously test LLMs' abilities in graph manipulation and dynamic reasoning. It includes a variety of graph encoding functions, with the addition of the adjacency matrix—a fundamental representation not explored in previous work. Compared to static graph property tasks, we find that SOTA LLMs' performance degrades notably on graph modification tasks, especially as the number of modifications increases. This decline is most pronounced when using the adjacency matrix encoding, highlighting the unique challenges posed by this encoding due to its dense and numerical nature.

To address this low performance, we explore and evaluate techniques aimed at improving LLM performance on graph modification tasks. We find that Chain-of-Thought (CoT) prompting can lead to performance increases for Claude 3.5 Sonnet and Llama 3.1 405B, yet we observe little to no performance gain in most cases where more CoT examples are included in the prompt. Across all baseline models and on multiple modification tasks, we find prompting the LLM to print intermediate graphs leads to notable performance gains. We call this technique **Modify-and-Print (MAP) prompting**, a simple yet effective technique where models are instructed to print the intermediate graph resulting from each modification step. MAP prompting significantly improves the models' ability to reason about the final graph over multiple modification steps. By explicitly generating the intermediate states, the models can better track changes and maintain accurate internal representations, leading to enhanced performance on the final tasks.

While we identify prompting techniques that improve the performance of LLMs on graph modification tasks, overall, LLMs are still not proficient in modifying graphs. The observed difficulties with graph modifications and adjacency matrix encodings underscore the need for improved models or training strategies that can handle dynamic, structured data more effectively. These results call for a shift in benchmarking practices toward tasks that require manipulation of graph data, thereby better aligning evaluations with real-world applications in dynamic networks and systems.

In summary, this work makes the following contributions:

1. Empirical Evidence of LLM Capabilities: We demonstrate that modern SOTA LLMs excel at basic graph property tasks across various encoding functions, challenging previous notions of their limitations.

2. Introduction of GraphModQA: We present a novel benchmark dataset designed to evaluate LLMs on graph modification tasks, providing a rigorous testbed for dynamic graph reasoning.

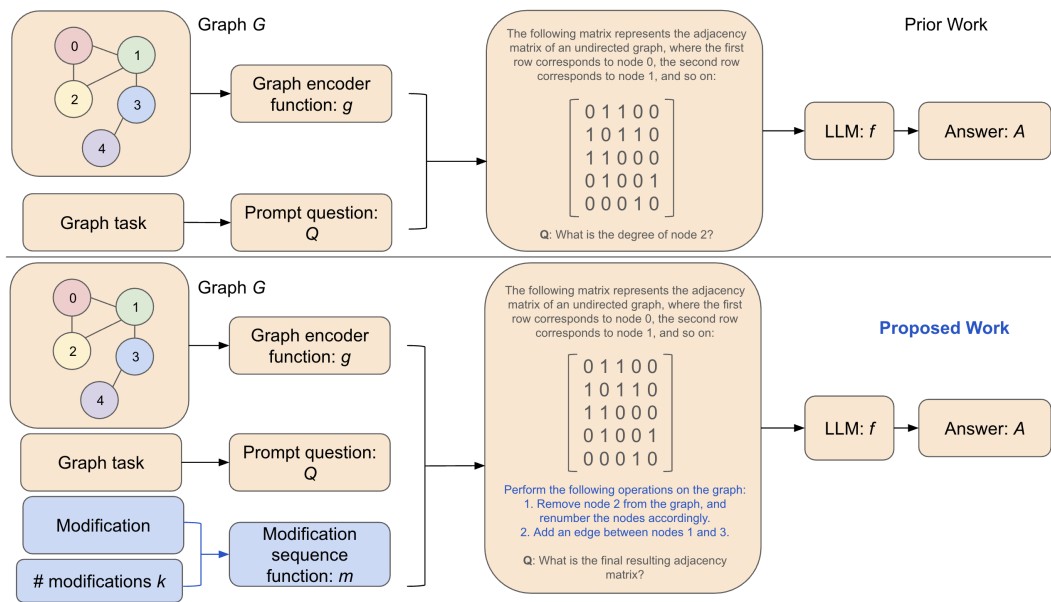

Figure 1: Previous work (Fatemi et al. (2023)) focus their effort on evaluating LLMs on graph property tasks (top), whereas this work focuses on graph modification tasks (bottom).

3. Analysis of LLM Performance on Graph Modifications: We reveal that SOTA LLMs experience significant performance degradation on graph modification tasks, especially with adjacency matrix encodings and increasing numbers of modifications.

4. Development of MAP Prompting Technique: We propose the Modify-and-Print (MAP) prompting method, which markedly improves LLM performance on dynamic graph tasks by instructing models to output intermediate adjacency matrices.

By addressing the challenges identified in this study, we aim to advance the development of LLMs capable of sophisticated reasoning over dynamic and structured data, thereby expanding their applicability in complex, real-world scenarios.

## 2 RELATED WORKS

**LLMs for graph reasoning:** Wang et al. (2024a) explores the capability of LLMs to tackle various graph-based tasks. This study evaluates tasks such as topological sort, maximum flow, and bipartite graph matching. Fatemi et al. (2023) delves into different methods for encoding graphs as text, with a particular focus on evaluating different encodings of graphs as text. This work builds upon Wang et al. (2024a) by introducing more interpretable, straightforward, and fundamental tasks, focusing on fundamental graph properties. The graph property tasks evaluated in Fatemi et al. (2023) include: **Node Count** (counting the total number of nodes in the graph), **Edge Count** (counting the total number of edges in the graph), **Node Degree** (calculating the degree of a given node), **Edge Existence** (determining if an edge exists between two nodes), **Connected Nodes** (identifying nodes connected to a given node), and **Cycle Check** (determining whether or not a cycle exists in the graph).

Both sets of tasks found in Wang et al. (2024a) and Fatemi et al. (2023) are limited to returning graph properties. In addition, Fatemi et al. (2023) use LLMs in the PaLM family as their benchmarks and concluded that LLMs perform poorly on these tasks. Our work evaluates SOTA LLMs on the graph property tasks defined in Fatemi et al. (2023), finding that modern LLMs strongly outperform PaLM 62B on these tasks. We then shift the focus to graph modification tasks, which potentially better evaluate the graph reasoning and manipulation capabilities of state-of-the-art LLMs.

Outside of Fatemi et al. (2023) and Wang et al. (2024a), there exists a small yet emerging body of work at the intersection of LLMs and graph reasoning. Perozzi et al. (2024) directly follows up on Fatemi et al. (2023) by utilizing and finetuning soft-token prompts to better encode graphs for LLMs, whereas this work aims to further investigate the inherit graph reasoning abilities of pre-trained LLMs. In this paper, following Fatemi et al. (2023) and Wang et al. (2024a), we freeze the parameters of the LLM, and the model operates in a black box setup, consuming and producing text without updating its parameters via any gradient-based optimization. Additionally, Zhang et al. (2023) addresses the challenges of solving spatial-temporal problems on dynamic graphs using LLMs, evaluating various LLMs' abilities to solve various spatio-temporal graph property tasks rather than their abilities to modify a graph manually. He et al. (2024) utilizes retrieval-augmented generation techniques to improve LLM performance on graph understanding and question answering. Guo et al. (2023) provides a broad empirical evaluation of LLMs' understanding of graph-structured data.

**LLMs for graph generation:** Yao et al. (2024) focuses on the generation of graphs from scratch by LLMs. It explores the potential of LLMs to create coherent and meaningful graph structures, whereas this work tests LLMs' abilities to modify existing ones. Wang et al. (2024b) examines how well LLMs can recall graph structures from text, emphasizing the accuracy of retrieving specific graph microstructures.

**Multi-step Reasoning Datasets:** Datasets that test multi-step or multi-hop reasoning abilities of LLMs are crucial for evaluating complex reasoning skills. Examples of such datasets include HotpotQA (Yang et al. (2018)), which involves answering questions that require synthesizing information from multiple Wikipedia articles, and ComplexWebQuestions (Talmor & Berant (2018)), which extends simple questions to multi-hop queries. These datasets challenge LLMs to perform intricate reasoning over multiple steps to arrive at a correct answer.

The babI dataset (Weston et al. (2015)) is another significant benchmark in this context, designed to test the multi-step reasoning capabilities of language models through a series of question-answering tasks that require the model to follow a chain of reasoning steps. Its significance lies in its structured and incremental approach to testing different types of reasoning, making it a valuable tool for assessing models' abilities to handle sequential logical operations. Our work aims to build a similar dataset to baBI, specifically targeting multi-step graph reasoning, in order to evaluate LLMs' abilities to perform reasoning over dynamic graph structures.

## 3 PRELIMINARIES

This section outlines our approach to measuring the graph reasoning abilities of LLMs, detailing the notation used, our evaluation methodology, and the construction of our datasets.

### 3.1 NOTATION

Let $f$ represent the interface function to a generative AI model, which processes high-dimensional discrete input tokens $W$ and produces output in the same token space. For this study, $f$ refers to a pre-trained Large Language Model (LLM). We define graphs as $G = (V, E)$, where $V$ is the set of nodes (vertices) and $E \subseteq (V \times V)$ is the set of edges connecting them.

### 3.2 GRAPH REASONING WITH LLMS

Following Fatemi et al. (2023), we evaluate an LLM's graph reasoning abilities by presenting it with a graph $G$ and a question $Q$. The LLM generates an answer $A$, which is compared to a ground-truth solution $S$. Fatemi et al. (2023) conducted their evaluation over a dataset $D$ of $(g(G), q(Q), S)$ triples . We define $g(G)$ as the graph encoding function, which can represent $G$ in various textual formats, such as an adjacency matrix, an incident list, or a descriptive format. Similarly, we define $q(Q)$ as the question rephrasing function, which can employ different prompting methods, including zero-shot prompting and chain-of-thought prompting (Wei et al. (2022)). The performance of the LLMs is evaluated by iterating over $D$, and calculating the proportion of answers $A = f(g(G), q(Q))$ that match with the corresponding ground-truth solution $S$.

### 3.3 GRAPHQA

Fatemi et al. (2023) developed GraphQA, a significant and comprehensive synthetic dataset comprising $(g(G), q(Q), S)$ triples. The questions in GraphQA target basic graph properties, such as counting nodes and edges, with answers formatted as simple integer counts or yes/no responses. The primary objective of Fatemi et al. (2023) was to explore the performance implications of various graph encoding functions $g(.)$, question rephrasing functions $q(.)$, and LLM architectures $f(.)$ on the GraphQA dataset.

## 4 GRAPHMODQA

In this section, we introduce **GraphModQA**, a novel dataset specifically designed to evaluate the graph modification capabilities of LLMs. We provide a detailed description of the graph generation process, the structure of the dataset, and the types of modifications included to rigorously test LLMs' abilities to manipulate graph structures.

### 4.1 GRAPH GENERATION

The foundation of GraphModQA lies in the diverse and robust generation of graph structures. Consistent with the methodologies outlined by Wang et al. (2024a) and Fatemi et al. (2023), we generate 250 undirected Erdős-Rényi (ER) graphs, where the total number of nodes in each graph, $n$, is sampled from a uniform distribution on a finite interval, and for each pair of nodes $(i, j)$, the probability $p$ that an edge exists between them is also sampled from a uniform distribution $U(0, 1)$. This diversity in graph structure is crucial for ensuring that the dataset comprehensively evaluates the LLMs' graph reasoning abilities across different graph configurations.

### 4.2 DATASET STRUCTURE

GraphModQA is constructed as a collection of $(g(G), m(M, k), q(Q), S)$ 4-tuples. Here, $G$ again represents the generated graph. We introduce $m(M, k)$ as the **modification sequence function**, which outputs a sequence of $k$ modifications of type $M$ to be performed on $G$. $Q$ denotes a final question on the resulting graph, and $S$ is the ground-truth solution to the question after all modifications have been applied. We illustrate this in Figure 1 and in Section A.9, which shows some example prompt inputs and model outputs. Additionally, we show the algorithms used to construct GraphModQA in Section A.3.

#### 4.2.1 GRAPH ENCODING FUNCTION $g(G)$

Similar to Fatemi et al. (2023), we define $g(G)$ as the graph encoding function, which represents the graph $G$ in a format suitable for input to the LLM. In GraphModQA, we utilize two encoding functions previously defined in Fatemi et al. (2023), namely the Incident List and Coauthorship encodings. In addition to these two encodings, we introduce the Adjacency Matrix encoding in this work and focus on it in the main sections of the paper due to its challenging nature for LLMs. The adjacency matrix is a matrix representation where each entry $A_{ij}$ indicates the presence (1) or absence (0) of an edge between nodes $i$ and $j$. Surprisingly, this encoding has yet to be explored as a graph encoding function in previous studies. Effectively manipulating adjacency matrices is important for LLMs because they are fundamental to many modern graph algorithms and applications. Adjacency matrices are widely used for storing and processing graph data in computational systems due to their suitability for matrix operations and compatibility with linear algebra-based techniques. Enabling LLMs to interpret and manipulate adjacency matrices extends their applicability to a broader range of real-world tasks in network analysis, computational biology, and machine learning for graphs, where adjacency matrices are a standard representation.

Additionally, the adjacency matrix provides a more challenging representation for the LLMs. This is not only because the adjacency matrix presents a dense numerical format that lacks the natural language cues of other encodings, or that the format forces models to reason on both the presence and absence of edges, but also because it relies on an implicit numbering scheme for nodes, where node identifiers correspond directly to the indices of the matrix. When modifications such as node removal occur, this implicit numbering becomes particularly challenging, as the nodes in the resulting

graph must be renumbered to maintain a contiguous matrix structure. For example, if an adjacency matrix represents nodes 0 to 4 and node 2 is removed, the third row and column are eliminated, and subsequent nodes are effectively renumbered—node 3 becomes node 2, node 4 becomes node 3. This renumbering adds an extra layer of complexity for the LLM to manage during reasoning and updates, increasing the difficulty of accurately interpreting and manipulating the graph.

### 4.2.2 MODIFICATION SEQUENCE FUNCTION $m(M, k)$

The modification sequence function $m(M, k)$ lists in text the sequence of $k$ modifications $m_1, m_2, ..., m_k$ to be performed on $G$, resulting in a final graph $G_k$. The intermediate modifications in each sequence must belong to the same modification type $M$, where these types include: **1) Add Edge** (instructing the model to add an edge between two sampled and unconnected nodes), **2) Remove Edge** (instructing the model to remove the existing edge between two sampled and connected nodes), **3) Add Node** (instructing the model to add a new node to the graph), **4) Remove Node** (instructing the model to remove an existing node from the graph, along with all its associated edges), and **5) Mix** (uniformly sampling one of the four previously defined modifications at each step $k$ in the sequence).

### 4.2.3 FINAL QUESTION Q AND QUESTION REPHRASING $q(Q)$

We ask the LLM to answer a final question $Q$ based on the final modified graph $G_k$. We include multiple graph property questions from Fatemi et al. (2023) and give additional details regarding these tasks in Section A.1. In addition to these property tasks, we introduce another final question, **Print Graph**, which requires the LLM to output the entire $G_k$ in the same format as the graph encoding function $g(G)$. This task is particularly challenging because it necessitates the model to accurately reconstruct and output the full graph structure after multiple modifications, demanding precise state tracking and a comprehensive internal representation.

We define q(Q) as the question rephrasing function, which can involve different prompting methods. In GraphModQA, we explore various prompting techniques to assess their impact on the models' performance, including **zero-shot prompting** (providing the question without any additional context or examples), **Chain-of-Thought (CoT) prompting with 1 to 3 examples** (including a list of examples that each demonstrate the reasoning process step-by-step), and **Modify-and-Print (MAP) prompting** (a novel prompting technique for graph modification tasks introduced in Section 5.3).

### 4.3 DATASET CONSTRUCTION

For each randomly generated input graph $G$, we define five components that can vary when constructing a single dataset entry:

- **Graph encoding function** $g()$: We use three encoding types: **Adjacency Matrix**, **Incident**, and **Coauthorship**.
- **Modification type** $M$: There are five modification types: **Add Edge**, **Remove Edge**, **Add Node**, **Remove Node**, and **Mix**.
- **Number of modifications** $k$: This ranges from 1 to 5.
- **Final question** $Q$: Five question types are used: **Node Count**, **Edge Count**, **Node Degree**, **Connected Nodes**, and **Print Graph**.
- **Question rephrasing function** $q()$: We employ five prompting methods: **zero-shot prompting**, **CoT prompting with one example**, **CoT with two examples**, **CoT with three examples**, and **MAP prompting**.

To illustrate the dataset construction process, we include Algorithm 6 in Section A.3, which describes how GraphModQA entries are generated. The algorithm assumes fixed graph encoding and question rephrasing functions. For each of the 250 initial graphs, the algorithm applies five rounds of modifications. In each round, five different types of modifications are performed, resulting in 5 modified versions of the graph per round. For each modified graph, five questions are posed from the predefined set of final questions Q. Each round builds upon the previous one, where each of the five modified graphs undergoes an additional modification. Thus, each initial graph contributes 5

modification rounds × 5 modifications × 5 questions = 125 entries to the dataset. To account for the three graph encoding functions and five question rephrasing methods, the total size of Graph-ModQA becomes: 250 graphs × 125 entries per graph × 3 encodings × 5 rephrasings = **468,750 unique examples**.

## 5 EXPERIMENTS

In this section, we summarize the results of our experiments. For each experiment, we evaluate using 4 SOTA LLMs: **GPT-4o mini**, **Llama 3.1 405B**, **Claude 3.5 Sonnet**, and **o1-mini**. We provide further implementation details of these experiments in Section A.2.

### 5.1 GRAPH PROPERTY TASKS

To establish a baseline and compare our findings with previous work, we evaluated the performance of state-of-the-art (SOTA) LLMs on basic graph property tasks similar to those presented in Fatemi et al. (2023). We follow Fatemi et al. (2023) by generating and evaluating on 500 ER graphs, where the size of each graph $n$ is drawn from $U(5, 20)$. We provide results obtained from our experiments with SOTA LLMs in the Appendix (see Tables 1 and 2). These tables clearly demonstrates that SOTA LLMs significantly outperform the PaLM models previously reported in Fatemi et al. (2023) on these basic graph property tasks. This substantial improvement highlights the advancements in LLM capabilities and sets the stage for our exploration of more complex graph reasoning tasks in the following sections.

### 5.2 GRAPH MODIFICATION TASKS

Building upon the baseline established with static graph property tasks, we investigated the performance of SOTA LLMs on the more challenging graph modification tasks introduced in Graph-ModQA. To evaluate the ability of LLMs to handle dynamic graph modifications, we utilized a dataset comprising 250 initial graphs where the size of each graph $n$ is drawn from $U(7, 20)$. For each of these graphs, using the **Adjacency Matrix** encoding, we applied 1 to 5 modifications for each of the five modification types—**Add Edge**, **Remove Edge**, **Add Node**, **Remove Node**, and **Mix**—resulting in multiple sets of modified graphs. After applying the specified modifications to each initial graph, we posed the **Print Graph** final question to the LLMs, instructing them to output the resulting modified graph in the form of an adjacency matrix. This comprehensive approach allows us to systematically evaluate the models' capabilities in maintaining and updating internal graph representations across varying levels of complexity. We illustrate the performance of each LLM in Figure 2, and we report the performance of each LLM on different graph encoders and final questions in the Appendix in Section A.6.

Our results indicate that across all modification types, models generally perform worse as the number of modifications increases, which suggests challenges in maintaining and updating an internal graph representation over multiple steps. Notably, the models perform the worst on the **Remove Node** and **Mix** modifications. The difficulty with the **Remove Node** modification can likely be attributed to the challenges associated with managing the adjacency matrix representation, where when a node is removed, not only must the corresponding row and column be deleted, but the indices of all subsequent nodes must be decremented to maintain the proper numbering scheme. In the **Mix** modifications, the models face the compounded challenge of handling a variety of modification types within a single sequence. The necessity to adapt to different operations—such as adding an edge in one step and removing a node in the next—requires flexible reasoning and robust state tracking, which current LLMs struggle to perform effectively with the adjacency matrix encoding. Overall, while Claude 3.5 Sonnet outperforms other models across the five modification types, o1-mini demonstrates superior performance on the two most challenging tasks, **Remove Node** and **Mix**, after a few modification steps. This suggests that o1-mini's internal reasoning capabilities become increasingly effective as the complexity of the modification sequence grows.

Overall, our findings indicate that while SOTA LLMs have made substantial progress in handling static graph property tasks, significant challenges remain in the context of dynamic graph modifications, especially when dealing with complex encodings like the adjacency matrix. These results emphasize the need for improved models and prompting techniques to enhance the graph reason-

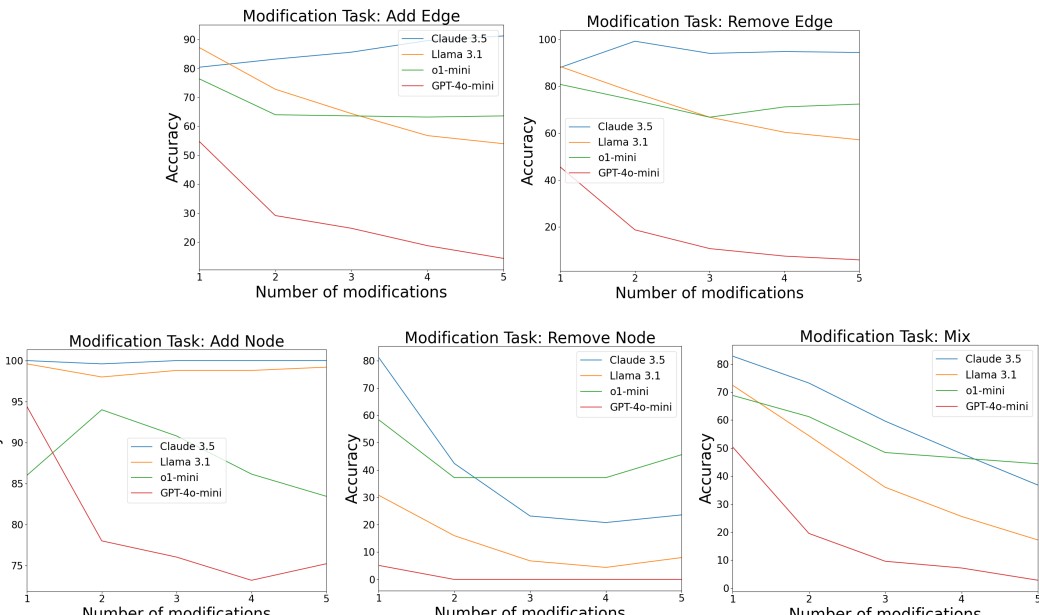

Figure 2: Results of all models on the **Print Graph** task, for each modification type, using the **Adjacency Matrix** encoding.

ing capabilities of LLMs in real-world applications involving dynamic and evolving networks. We include a detailed analysis on the types of errors these models make on the **Print Graph** task, as well as the frequencies of these errors, in Section A.8. In subsequent sections, we explore potential methods to improve performance.

## 5.3 IN-CONTEXT LEARNING

In this section, we explore potential methods for increasing the performance of LLMs on graph modification tasks, focusing on the adjacency matrix encoder and the **Print Graph** task. We track the performance of various in-context learning methods across 1 to 5 modification steps for the same 250 graphs, and compare the performance of these methods to the zero-shot performance reported in the previous section. We show the results for o1-mini in Figure 3, and results for the other LLMs can be found in Section A.4.

### 5.3.1 CHAIN-OF-THOUGHT PROMPTING

Chain-of-thought (CoT) prompting (Wei et al. (2022)) is a technique that encourages the model to generate intermediate reasoning steps before producing the final answer. By providing examples of detailed reasoning in the prompt, the model is guided to follow a similar process when answering new questions. In our experiments, we evaluated the impact of including 1, 2, and 3 CoT examples in the prompt on the models' performance. The models differed crucially in how much they were helped by CoT prompting. For Claude 3.5 Sonnet and Llama 3.1 405B, we observed in Figures 4 and 5 respectively that CoT prompting generally helps boost performance across all five modification types. In contrast, for GPT-4o mini, we did not observe significant changes in performance with CoT prompting, as shown in Figure 6. For all models, CoT performance remained relatively consistent regardless of the number of examples included, indicates that they may not be leveraging the additional reasoning steps provided in the prompt to enhance its performance on these tasks.

Interestingly, with the o1-mini model, we observed a large drop in performance when using CoT prompting (Figure 3). In general, the model performed worse with CoT examples compared to zero-shot prompting across all modification types. This decline in performance is likely due to the fact that o1-mini reasons internally, and external CoT prompting does not complement its internal reasoning processes.

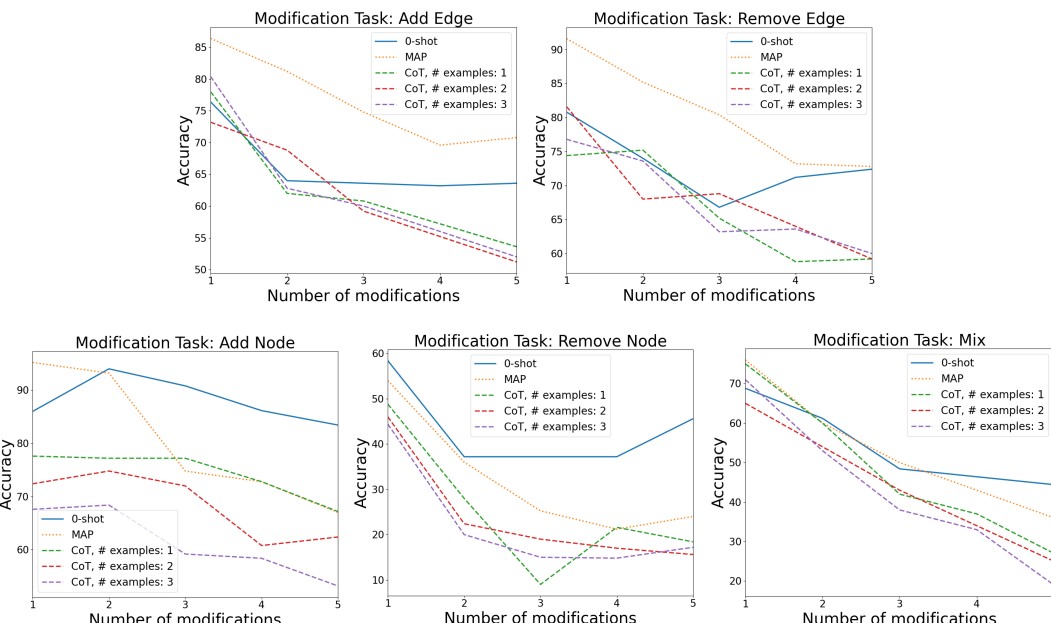

Figure 3: o1-mini In Context Learning Results.

### 5.3.2 MODIFY-AND-PRINT (MAP) PROMPTING

We introduce the **Modify-and-Print (MAP) prompting** technique as a simple yet effective method to improve model performance on graph modification tasks. In MAP prompting, we instruct the model to output every intermediate graph after each modification. By requiring the model to explicitly generate the intermediate adjacency matrices, we aim to help it maintain a clear internal representation of the graph as it undergoes changes. We illustrate MAP prompting in Section A.9.

We find that MAP prompting performs especially well on edge-related tasks (**Add Edge** and **Remove Edge**). This is evident for the o1-mini and Claude 3.5 Sonnet models in Figures 3 and 4, where MAP prompting consistently achieves higher accuracy than zero-shot and CoT prompting. The improvement is especially pronounced with the o1-mini model, where there is a large gap between MAP prompting and the other methods. This suggests that MAP prompting effectively supports the model's reasoning process by reinforcing state tracking through explicit output of intermediate graphs. On the other modifications—**Add Node**, **Remove Node**, and **Mix**—MAP prompting tends to remain competitive with CoT prompting.

An interesting observation emerges when examining the performance of MAP prompting on the first modification step. Intuitively, MAP prompting should perform near-identically to zero-shot prompting when only one modification is applied, as there is only one intermediate modification, which is the final answer. However, we observe that MAP prompting greatly outperforms zero-shot prompting even on the first modification step. This indicates that the presence of the instruction to output intermediate graphs has a significant positive effect on the models' performance. Furthermore, this suggests that MAP prompting not only aids in state tracking but also likely enhances the models' attention to the modification process, leading to more accurate outputs.

Overall, MAP prompting demonstrates its potential as a powerful technique to improve LLM performance on dynamic graph reasoning tasks. By encouraging explicit generation of intermediate states, it helps models navigate complex sequences of modifications, especially in tasks involving edge additions and removals. This finding underscores the importance of prompting strategies that align closely with the reasoning demands of the task.

## 6 CONCLUSION

In this paper, we have explored the graph reasoning capabilities of state-of-the-art large language models (LLMs) by introducing **GraphModQA**, a novel benchmark designed to assess models on dynamic graph modification tasks. Our findings reveal that while modern LLMs excel at basic graph property tasks—a significant improvement over previous generations—they exhibit notable performance degradation when tasked with modifying graphs, especially as the number of modifications increases. This decline is most pronounced when using the newly-explored **Adjacency Matrix** encoding, highlighting the challenges LLMs face in interpreting and manipulating dense numerical representations that require precise state tracking, node renumbering, and the recognition of both the presence and absence of edges.

To address these challenges, we investigated the effectiveness of in-context learning strategies, including **Chain-of-Thought (CoT) prompting** and our proposed **Modify-and-Print (MAP) prompting** technique. We found that both CoT and MAP prompting can significantly improve model performance, but their effectiveness varies depending on the task and the model. CoT prompting aids models by providing guided reasoning steps, which is particularly beneficial for models like Claude 3.5 and Llama 3.1 in handling complex decision-making processes. MAP prompting enhances performance by requiring models to explicitly generate intermediate graph states, thereby aiding in state tracking and manipulation tasks—this was especially effective across all models in edge addition and removal modifications. These findings suggest that leveraging the appropriate prompting technique can help overcome specific challenges in dynamic graph reasoning. Our work highlights the importance of tailored prompting strategies and calls for further research into methods that enhance LLMs' abilities to process and reason about dynamic graph structures in various contexts.

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

# A APPENDIX

## A.1 FINAL QUESTIONS

After applying the sequence of $k$ modifications to the initial graph $G$, we pose a final question $Q$ to the LLM based on the resulting modified graph $G_k$. The final questions are designed to assess the model's understanding and reasoning about the graph's properties and structure. The following are the types of final questions included in our evaluation:

- **Node Count**: *Calculate the total number of nodes in the modified graph $G_k$*. This question evaluates the model's ability to accurately track the addition or removal of nodes throughout the modification sequence.

- **Edge Count**: *Calculate the total number of edges present in the modified graph $G_k$* This requires the model to account for all edge additions and deletions, testing its capability to maintain an updated edge set.

- **Node Degree**: *Report the degree of a uniformly sampled node in the modified graph $G_k$.* The degree of a node is the number of edges incident to it. This question assesses the model's understanding of local graph topology and its ability to compute node-specific properties after modifications.

- **Connected Nodes**: *List all nodes that are directly connected to a uniformly sampled node in the modified graph $G_k$.* This task tests the model's ability to identify and retrieve adjacency information for a given node.

- **Print Graph**: *Output the entire modified graph $G_k$ in the same format as the graph encoding function $g(G)$.* This question is particularly challenging as it requires the model to reconstruct the complete structure of the graph after all modifications, demanding precise state tracking and comprehensive internal representation.

These final questions cover both global properties (e.g., node and edge counts) and local properties (e.g., node degree, connected nodes), as well as the ability to reproduce the full graph structure. By including a variety of question types, we aim to thoroughly evaluate the LLMs' proficiency in understanding, manipulating, and reasoning about graphs after dynamic changes.

## A.2 IMPLEMENTATION DETAILS

For our experiments, we evaluated four SOTA LLMs, o1-mini, GPT 4o-mini, Claude 3.5 Sonnet, and Llama 3.1 405B, using the OpenAI, Anthropic, and Fireworks AI APIs. We set the decoding temperature of all models to zero. We used the NetworkX library Hagberg et al. (2008) to generate all ER, Star, Path, Complete and Empty graphs, as well as the solutions to each final question. For each of the 250 ER input graphs from Section 5.2, the total number of nodes in each graph, $n$, is is drawn from $U(7, 20)$, and for each pair of nodes $(i, j)$, the probability $p$ that an edge exists between them is also sampled from a uniform distribution $U(0, 1)$. We used Algorithm 6 to generate the entire GraphModQA dataset, resulting in 468,750 unique examples generated from the 250 input graphs. We restricted our evaluation of all four models to these examples due to monetary costs, and encourage future research to expand this dataset for further evaluation.

### A.3 DATASET GENERATION ALGORITHM

In this section, we provide the pseudocode for the algorithms necessary for generating the Graph-ModQA dataset. Algorithms 1, 2, 3, 4, and 5 describe the generation behind individual modification instructions, and 6 describes the dataset generation process for GraphModQA. Regarding Algorithm 4, when evaluating the LLMs on final questions that used the **Adjacency Matrix** encoding and involved **Remove Node** modifications, we found that their performance was nearly zero until we appended the phrase *"and renumber the nodes accordingly"* to *"Remove node $v$ from the graph"*. This highlights the importance of providing explicit instructions to LLMs when tasks involve implicit node numbering schemes, as it ensures they correctly update and interpret the modified graph representations.

---

**Algorithm 1** ADDEDGE

---

**Require:** Graph $G$
**Ensure:** Modified Graph $G'$
 1: $G' \leftarrow G$
 2: $(i, j) \sim \mathcal{U}(V_{G'} \times V_{G'} \setminus E_{G'})$
 3: $E_{G'} \leftarrow E_{G'} \cup \{(i, j)\}$
 4: **return** $G'$, *"Add an edge between nodes $i$ and $j$."*

---

**Algorithm 2** REMOVEEDGE

---

**Require:** Graph $G$
**Ensure:** Modified Graph $G'$
 1: $G' \leftarrow G$
 2: $(i, j) \sim \mathcal{U}(E_{G'})$
 3: $E_{G'} \leftarrow E_{G'} \setminus \{(i, j)\}$
 4: **return** $G'$, *"Remove the edge between nodes $i$ and $j$."*

---

**Algorithm 3** ADDNODE

---

**Require:** Graph $G$
**Ensure:** Modified Graph $G'$
 1: $G' \leftarrow G$
 2: $V_{G'} \leftarrow V_{G'} \cup \{v\}, E_{G'} \leftarrow E_{G'}$
 3: **return** $G'$, *"Add a node $v$ to the graph."*

---

**Algorithm 4** REMOVENODE

---

**Require:** Graph $G$
**Ensure:** Modified Graph $G'$
 1: $G' \leftarrow G$
 2: $v \sim \mathcal{U}(V_{G'})$
 3: $V_{G'} \leftarrow V_{G'} \setminus \{v\}, E_{G'} \leftarrow E_{G'} \setminus \{(v, u) \mid u \in V_{G'}\}$
 4: **return** $G'$, *"Remove node $v$ from the graph."*

---

**Algorithm 5** MIX

---

**Require:** Graph $G$
**Ensure:** Modified Graph $G'$
 1: $G' \leftarrow G$
 2: MODIFICATION $\sim \mathcal{U}(\{$ADDEDGE, REMOVEEDGE, ADDNODE, REMOVENODE$\})$
 3: **return** MODIFICATION($G'$)

---

---

**Algorithm 6** ConstructGraphModQA

---

**Require:** Number of graphs to generate $N$
**Ensure:** Dataset $D$ containing multi-step tasks for all final queries and $k$ values
1: Initialize an empty dataset $D$
2: Define the set of possible final questions $\mathcal{Q} = \{$Node Count, Edge Count, Node Degree, Connected Nodes, Print Graph$\}$
3: Define the maximum number of modifications $k_{max} = 5$
4: Define $V_G$ as the set of nodes in any graph $G$, and $E_G$ as the set of edges in any graph $G$
5: **for** $i = 1$ to $N$ **do**
6:     Sample $n \sim \mathcal{U}(7, 20)$
7:     Generate an undirected Erdős-Rényi graph $G = (V, E)$ with $|V| = n$ and sample edge probability $p \sim \mathcal{U}(0, 1)$
8:     Initialize graphs $G_{AE}, G_{RE}, G_{AN}, G_{RN}, G_{MX} \leftarrow G$
9:     Initialize $M_{AE}, M_{RE}, M_{AN}, M_{RN}, M_{MX} \leftarrow []$
10:     **for** $k = 1$ to $k_{max}$ **do**
11:       $G_{AE}, m_{AE} \leftarrow$ ADDEDGE$(G_{AE})$ 1
12:       $M_{AE} \leftarrow M_{AE} \parallel m_{AE}$
13:       $G_{RE}, m_{RE} \leftarrow$ REMOVEEDGE$(G_{RE})$ 2
14:       $M_{RE} \leftarrow M_{RE} \parallel m_{RE}$
15:       $G_{AN}, m_{AN} \leftarrow$ ADDNODE$(G_{AN})$ 3
16:       $M_{AN} \leftarrow M_{AN} \parallel m_{AN}$
17:       $G_{RN}, m_{RN} \leftarrow$ REMOVENODE$(G_{RN})$ 4
18:       $M_{RN} \leftarrow M_{RN} \parallel m_{RN}$
19:       $G_{MX}, m_{MX} \leftarrow$ MIX$(G_{MX})$ 5
20:       $M_{MX} \leftarrow M_{MX} \parallel m_{MX}$
21:       $Mods = \{(G_{AE}, M_{AE}), (G_{RE}, M_{RE}), (G_{AN}, M_{AN}), (G_{RN}, M_{RN}), (G_{MX}, M_{MX})\}$
22:       **for** $Q \in \mathcal{Q}$ **do**
23:         **for** $(G_{Mod}, M_{Mod}) \in Mods$ **do**
24:           **if** $Q =$ Node Count **then**
25:             $S \leftarrow |V_{G_{Mod}}|$
26:           **else if** $Q =$ Edge Count **then**
27:             $S \leftarrow |E_{G_{Mod}}|$
28:           **else if** $Q =$ Node Degree **then**
29:             $v \sim \mathcal{U}(V_{G_{Mod}})$
30:             $S \leftarrow |\{u \in V_{G_{Mod}} \mid (v, u) \in E_{G_{Mod}}\}|$
31:           **else if** $Q =$ Connected Nodes **then**
32:             $v \sim \mathcal{U}(V_{G_{Mod}})$
33:             $S \leftarrow \{u \in V_{G_{Mod}} \mid (v, u) \in E_{G_{Mod}}\}$
34:           **else if** $Q =$ Print Graph **then**
35:             $S \leftarrow G_{Mod}$
36:           **end if**
37:           $D \leftarrow D \cup (G, M_{Mod}, Q, S)$
38:         **end for**
39:       **end for**
40:     **end for**
41: **end for**
42: **return** $D$

---

## A.4 FURTHER IN CONTEXT LEARNING RESULTS

In this section we show the in context learning results achieved by Claude 3.5 Sonnet, Llama 3.1 405B, and GPT-4o mini on the **Print Graph** task using the **Adjacency Matrix** encoder.

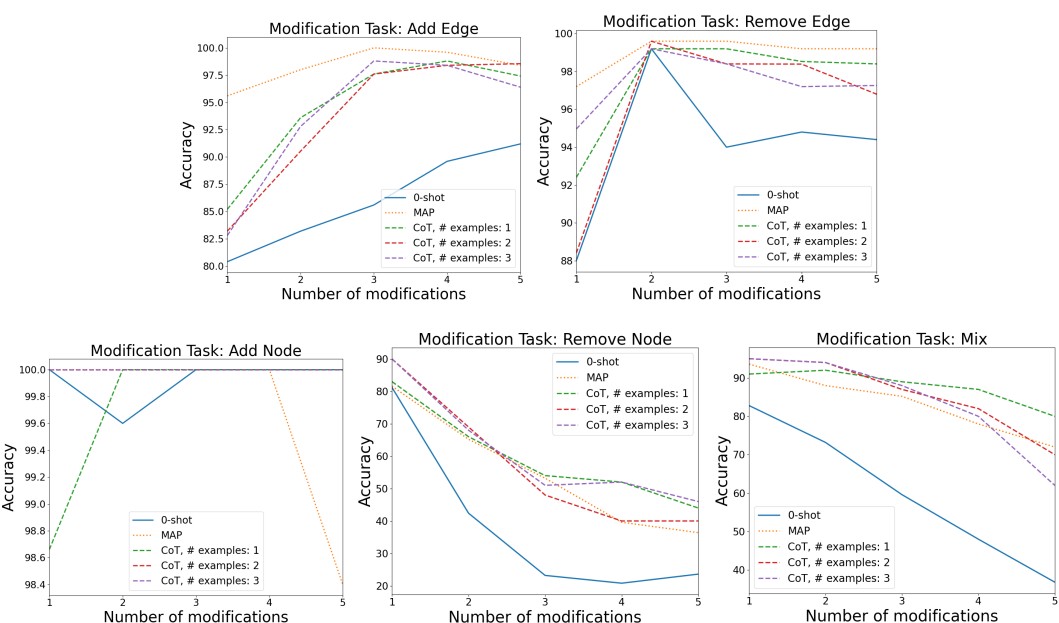

Figure 4: Claude 3.5 Sonnet In Context Learning Results.

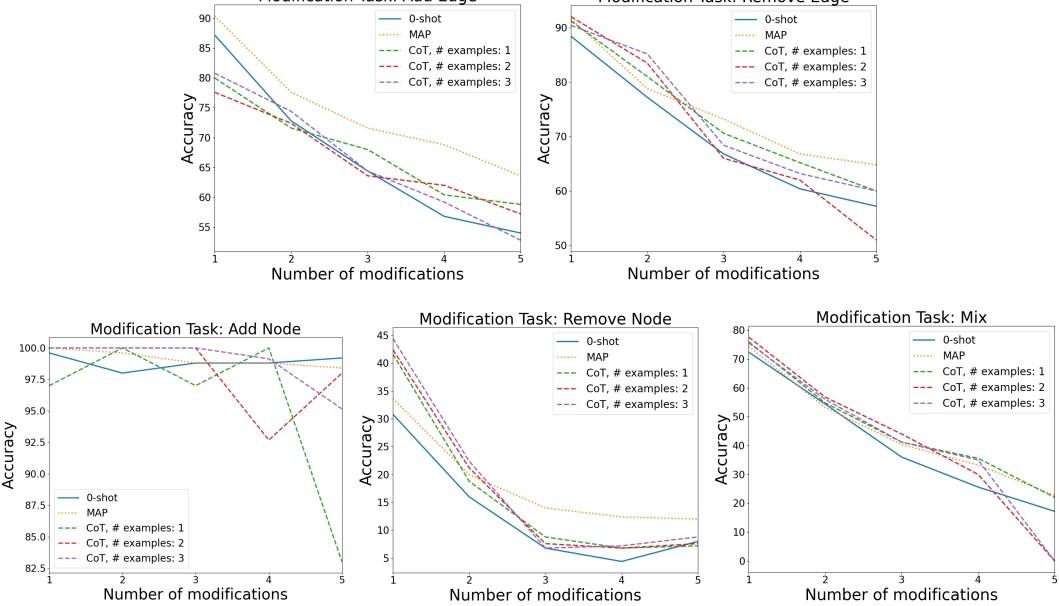

Figure 5: Llama 3.1 405B In Context Learning Results.

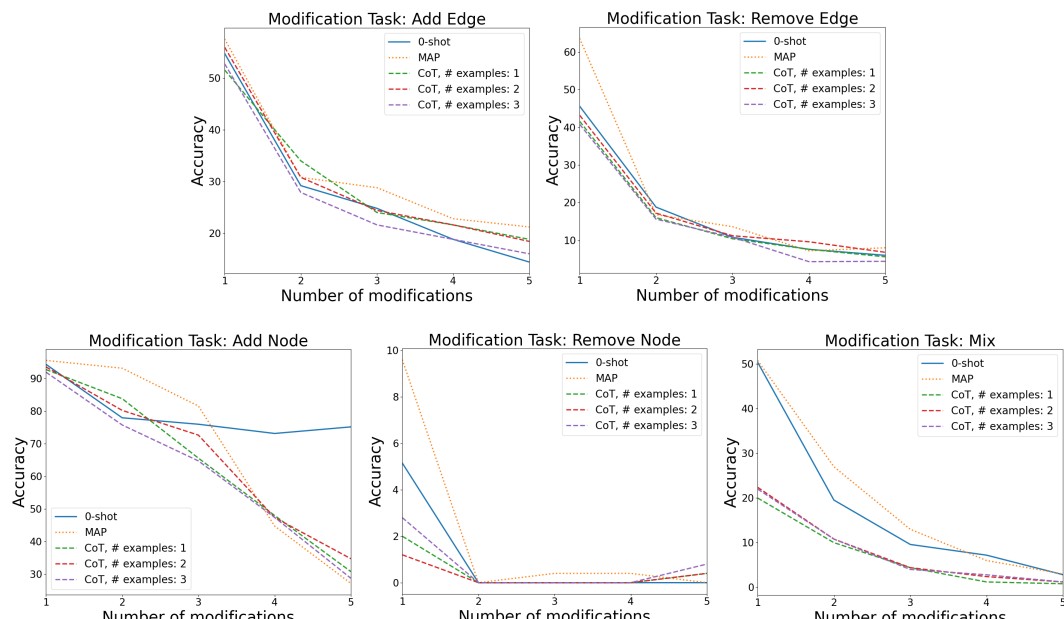

Figure 6: GPT-4o mini In Context Learning Results.

## A.5 GRAPH TYPES & THE PRESERVATION OF GRAPH STRUCTURE

In this section, we perform additional experiments to investigate the ability of Llama 3.1 405B, Claude 3.5 Sonnet, and o1-mini to print the adjacency matrix across three modification steps of different graph types, including: **1) star graphs**, **2) path graphs**, **3) complete graphs**, and **4) empty graphs**. We evaluate each LLM on 250 graphs of each graph type.

Figures 7, 8, and 9 show the varying levels of strength each model exhibits on graph modification for these more structured graph types. Interestingly, across all three models, we notice notable dropoffs in performance at varying modification steps for the **Add Node** modification on the complete graph. This drop in performance is most notable in the o1-mini model, which exhibits poor performance across all graph types.

To explore this, for each graph type, we analyze the percentage of errors o1-mini makes that involves connecting the newly added node to either the central node for star graphs, the final node in the path (at the bottom row of the adjacency matrix) for path graphs, all existing nodes for complete graphs, and any node for empty graphs. We show these results in Figure 10.

We observe that o1-mini has an extremely high intrinsic bias to connect the incoming node, and in this way, o1-mini attempts to **preserve the underlying structure of the input graph**.

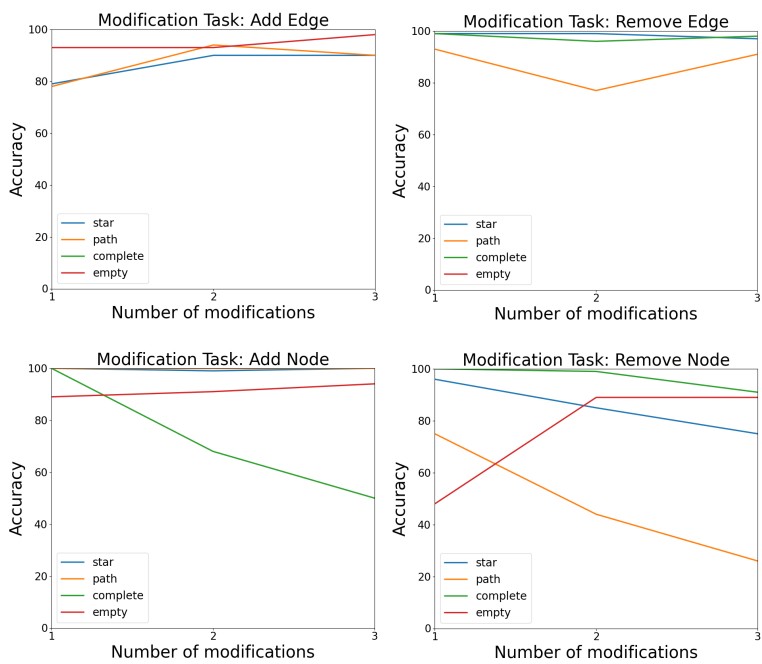

Figure 7: Claude 3.5 Sonnet performance on different graph types.

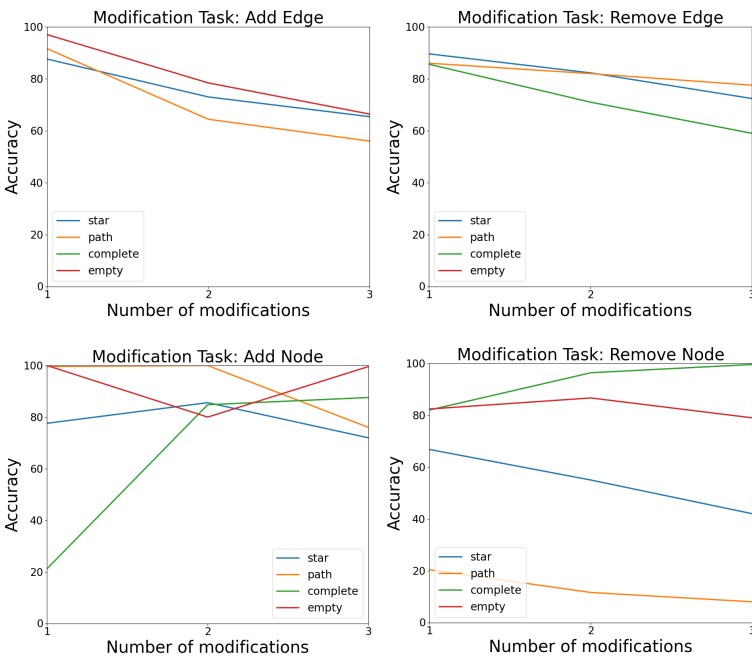

Figure 8: Llama 3.1 405B performance on different graph types.

## A.6 RESULTS ON VARYING FINAL QUESTIONS AND GRAPH ENCODERS

For the **Adjacency Matrix** encoding, in addition to the **Print Graph** question, we evaluated model performance on other final questions, including **Node Count**, **Edge Count**, **Node Degree**, and **Connected Nodes**. Detailed results for these tasks are provided in Figures 11, 14, 17, and 20 respectively. Our analysis reveals that models consistently perform poorly on the **Print Graph** task when compared to other graph property tasks. This finding is significant because it shows that maintaining the modified structure itself was approximately as challenging as computing quantities

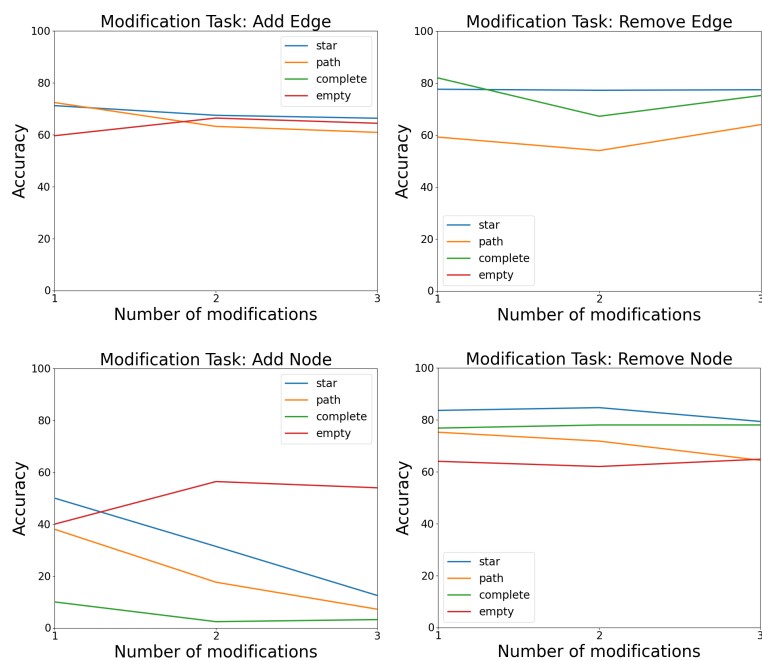

Figure 9: o1-mini performance on different graph types.

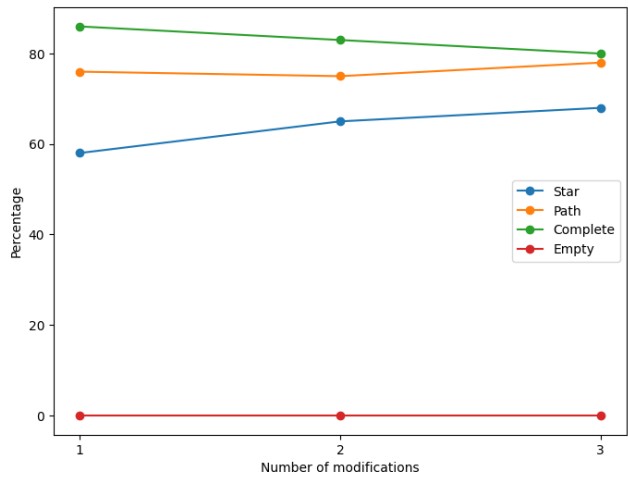

Figure 10: Percentage of total errors that involve o1-mini connecting the newly added node.

derived from it: outputting the entire adjacency matrix requires managing the structured data it contains, and it suggests that a key part of the challenge in this problem comes as much from this form of data maintenance as from computations performed on it.

Furthermore, we explored the impact of different graph encoding functions on model performance. Besides the **Adjacency Matrix** encoding, we included the **Incident** and **Coauthorship** encodings from Fatemi et al. (2023) in our experiments. Results for these encodings are presented in Figures 12, 13, 15, 16, 18, 19, 21, 22, 23, and 24. We observed that models generally perform substantially worse on the **Adjacency Matrix** encoding compared to the other two encodings. With very few exceptions, the performance of the models decreases as the number of modifications increases, highlighting the challenges LLMs face when dealing with both dense numerical representations that lack explicit linguistic cues and with increasingly complex sequences of modifications. Therefore, future benchmarking efforts for graph reasoning should focus on the adjacency matrix encoder to

better assess and improve LLMs' abilities to handle complex, structured graph representations. Below, we analyze the performance of all models on each of the final question types:

### A.6.1 NODE COUNT

Across all three encodings, nearly all LLMs achieve close to 100% accuracy across the five modification steps, except GPT 4o-mini. o1-mini demonstrates slight drops in performance on all modification types compared to Claude 3.5 Sonnet and Llama 3.1 405B on the **Adjacency Matrix** encoding. This observation follows from Table 2, which also indicates that even in the static case, o1-mini lags slightly behind both Claude 3.5 Sonnet and Llama 3.1 405B on counting the number of nodes in an adjacency matrix.

### A.6.2 EDGE COUNT

o1-mini consistently outperforms all other models across the three encodings, aligning with the trends observed in Table 1. Among the encodings, the **Adjacency Matrix** encoding is the most challenging for all models, likely because its dense representation makes it harder for LLMs to infer and count edge relationships directly. Conversely, models perform best on the **Incident** encoding, as it explicitly represents the connections between nodes with numerical node ID. This structure simplifies edge tracking and counting for the models.

### A.6.3 NODE DEGREE

All models maintain nearly 100% accuracy on the **Incident** and **Coauthorship** encodings, except GPT 4o-mini, which lags slightly. The adjacency matrix encoding presents the most difficulty, especially on Remove Node modifications. This is unsurprising because removing a node in an adjacency matrix requires adjustments across both rows and columns, increasing the likelihood of errors. On this encoding, o1-mini again outperforms others for all modification types except Add Node. The **Adjacency Matrix** results underscore that remove node is inherently a more error-prone operation due to the renumbering and recalibration of indices. Interestingly, Claude 3.5 Sonnet's performance increases slightly on the Add Node modification as the number of modifications increase.

### A.6.4 CONNECTED NODES

The Connected Nodes task mirrors the patterns found in Node Degree. o1-mini outperforms all other models on **Adjacency Matrix** encoding, the most challenging format. As with Node Degree, the Remove Node modification introduces the most notable performance drop for all models. Llama 3.1 405B shows slight improvement in accuracy for Add Node modifications as the number of modifications increases.

### A.6.5 PRINT GRAPH

Performance varies significantly across encodings. Models perform much better on the **Incident** and **Coauthorship** encodings than on the **Adjacency Matrix**, with **Incident** again being the easiest to process.

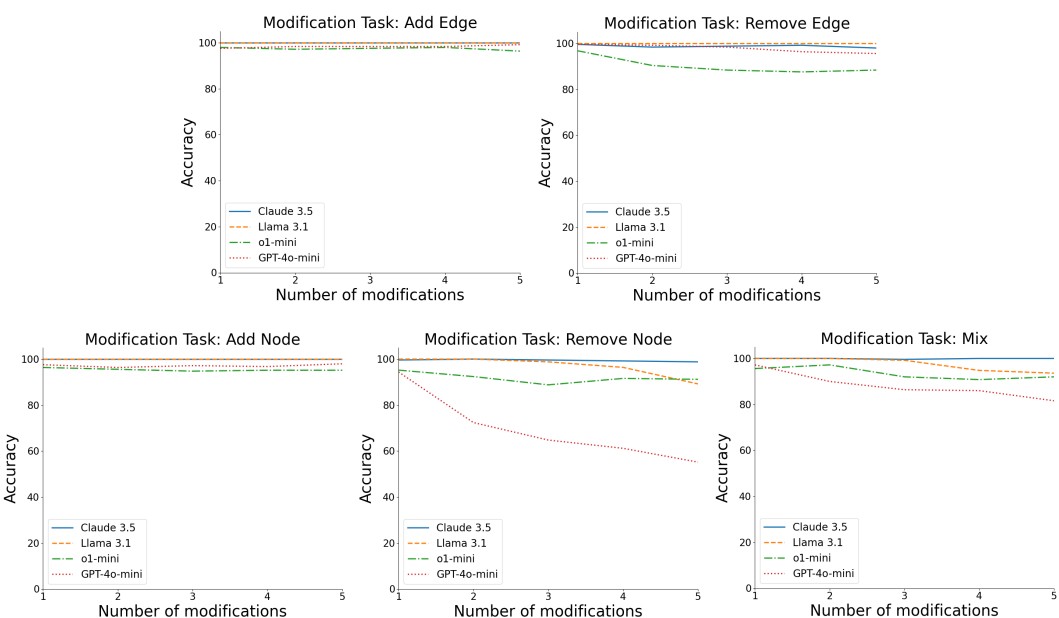

Figure 11: Node Count, Adjacency Matrix.

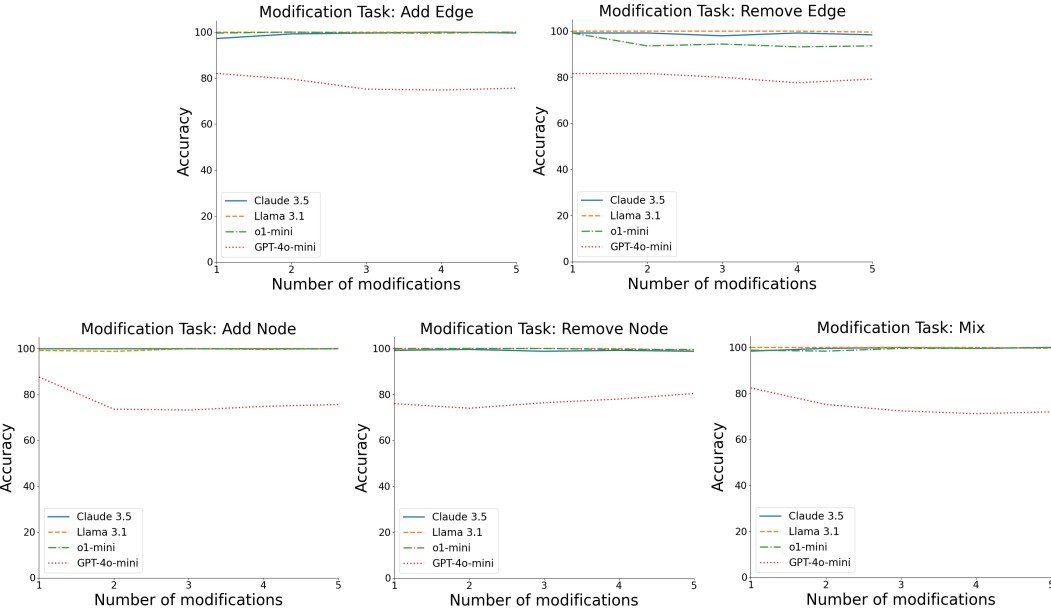

Figure 12: Node Count, Coauthorship.

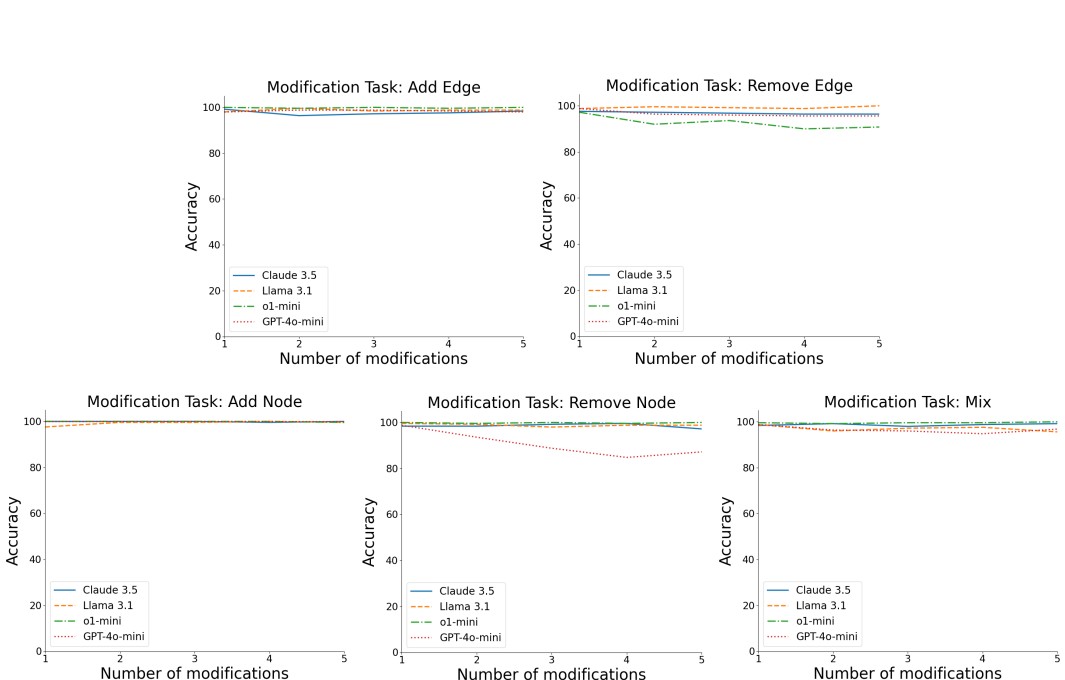

Figure 13: Node Count, Incident List.

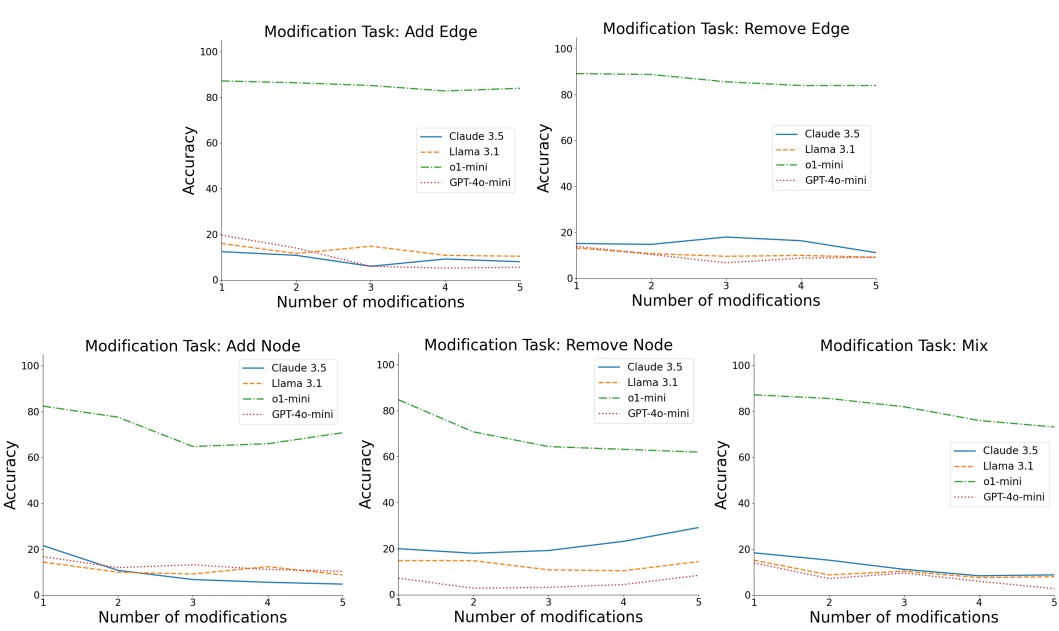

Figure 14: Edge Count, Adjacency Matrix.

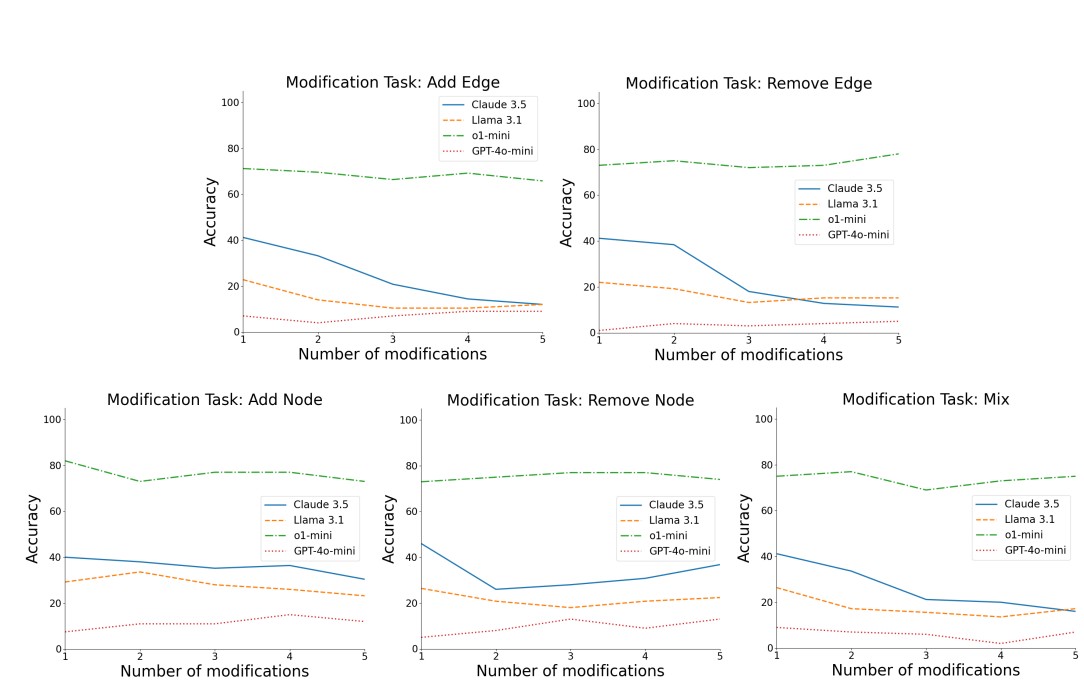

Figure 15: Edge Count, Coauthorship.

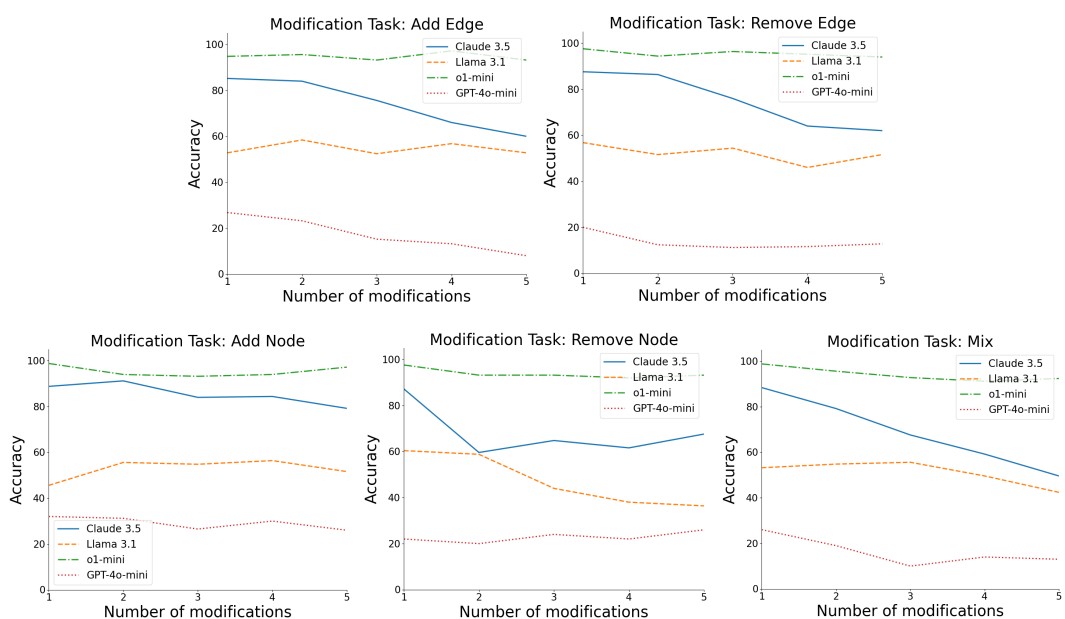

Figure 16: Edge Count, Incident List.

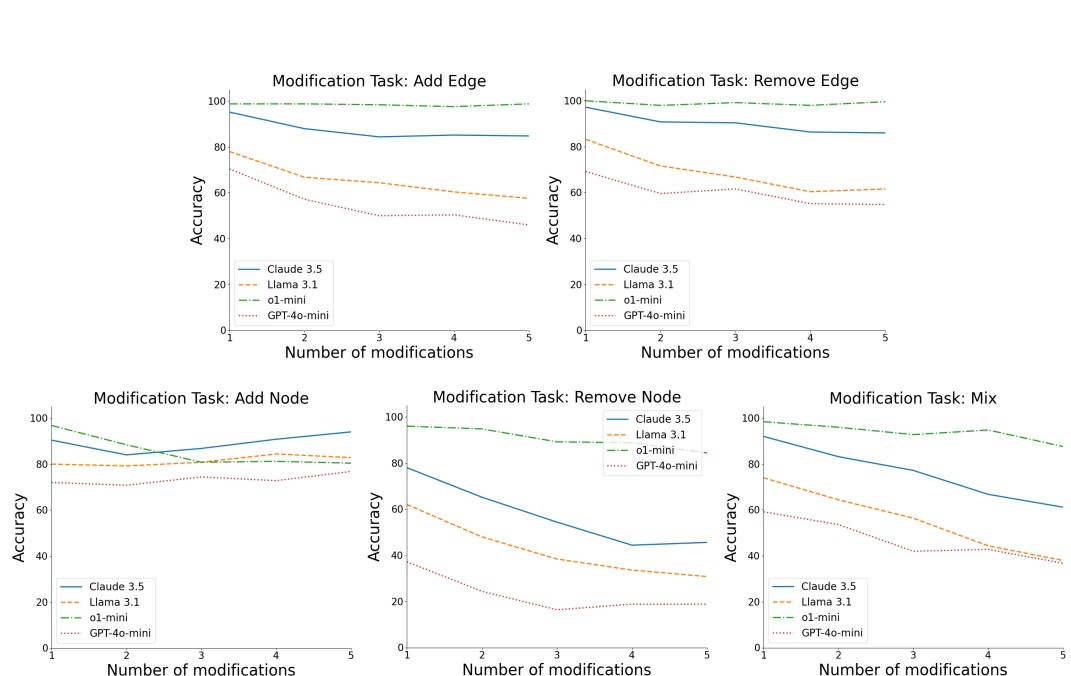

Figure 17: Node Degree, Adjacency Matrix.

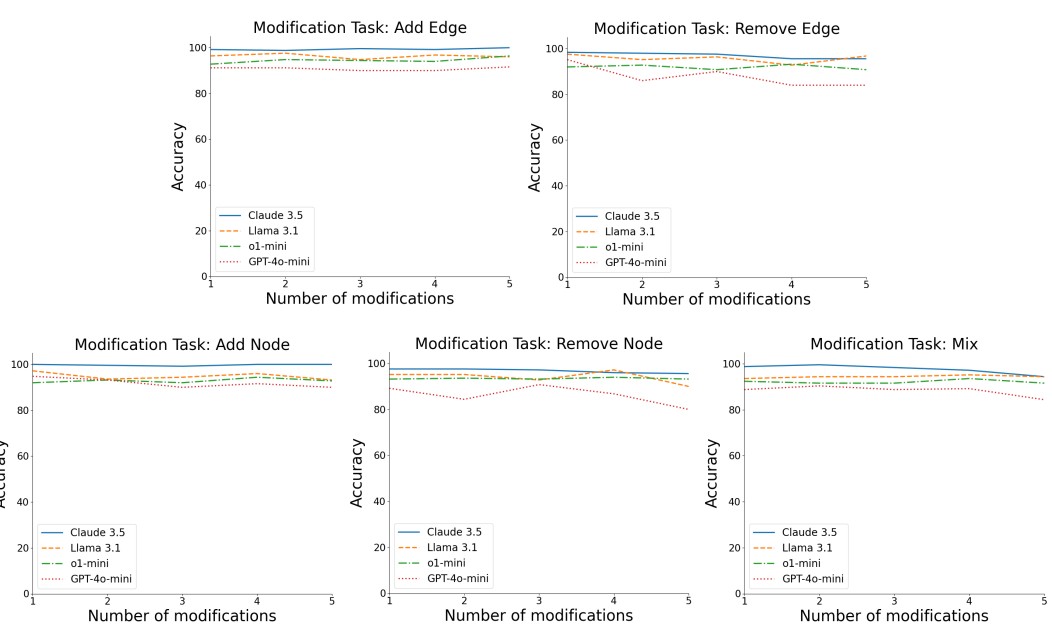

Figure 18: Node Degree, Coauthorship.

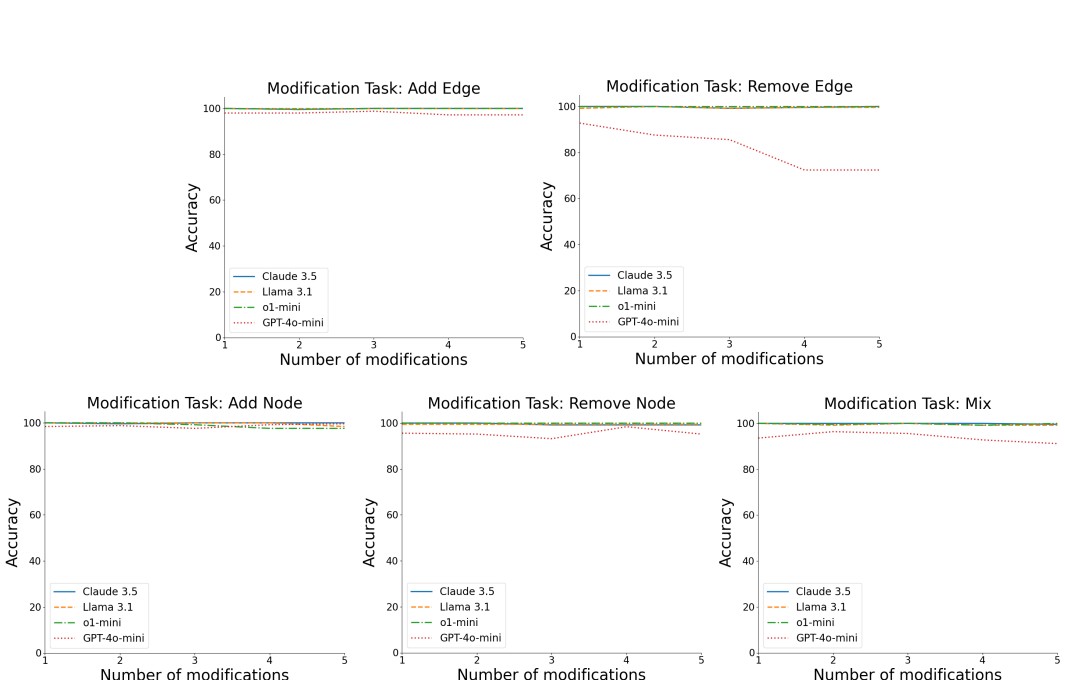

Figure 19: Node Degree, Incident List.

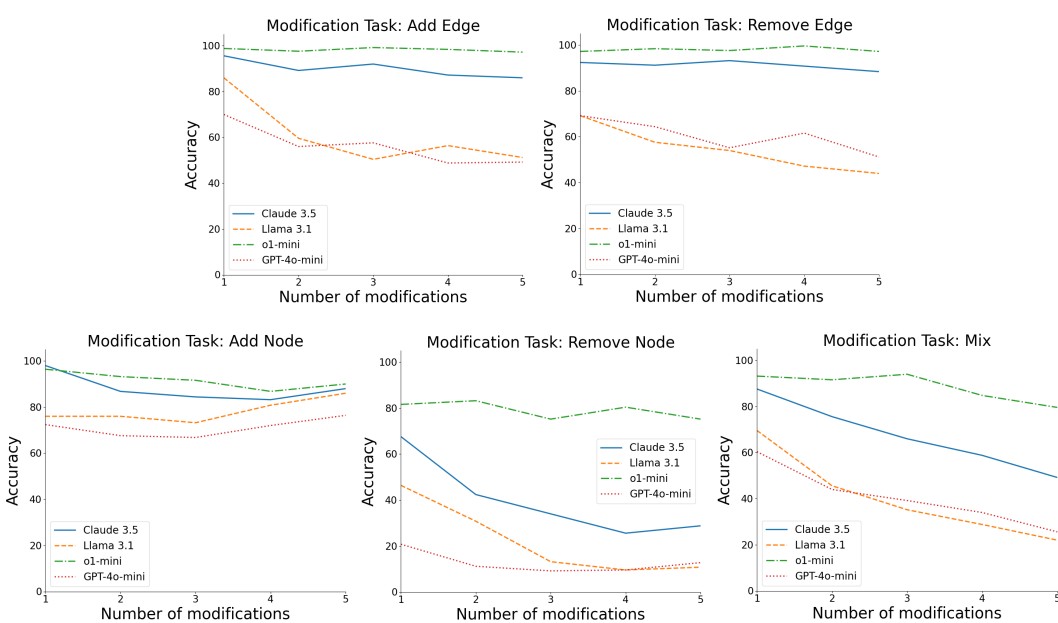

Figure 20: Connected Nodes, Adjacency Matrix.

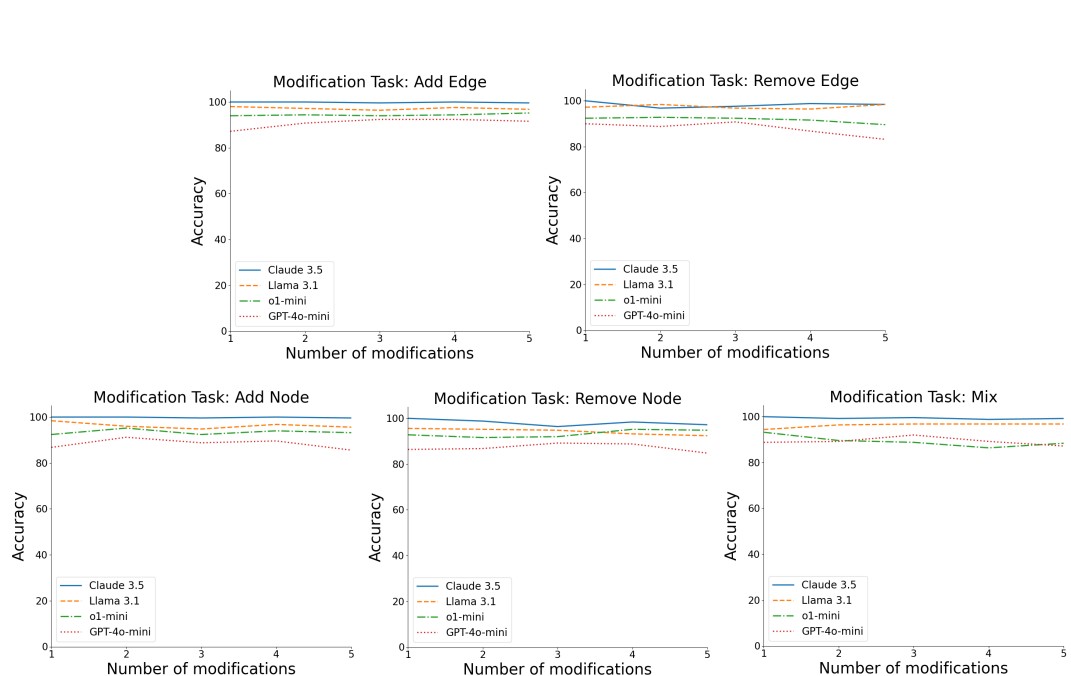

Figure 21: Connected Nodes, Coauthorship.

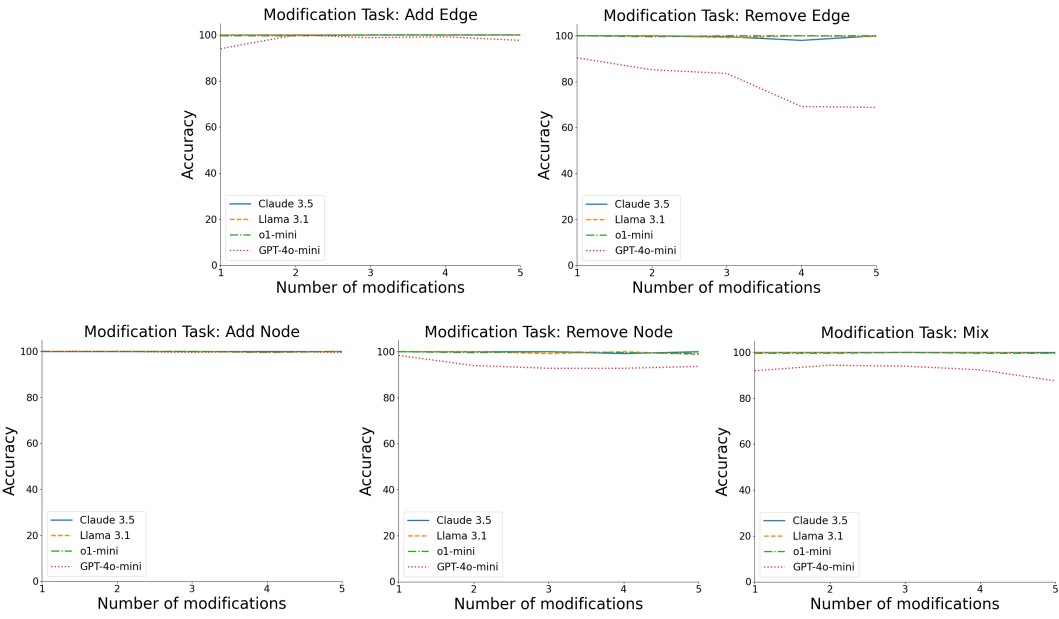

Figure 22: Connected Nodes, Incident List.

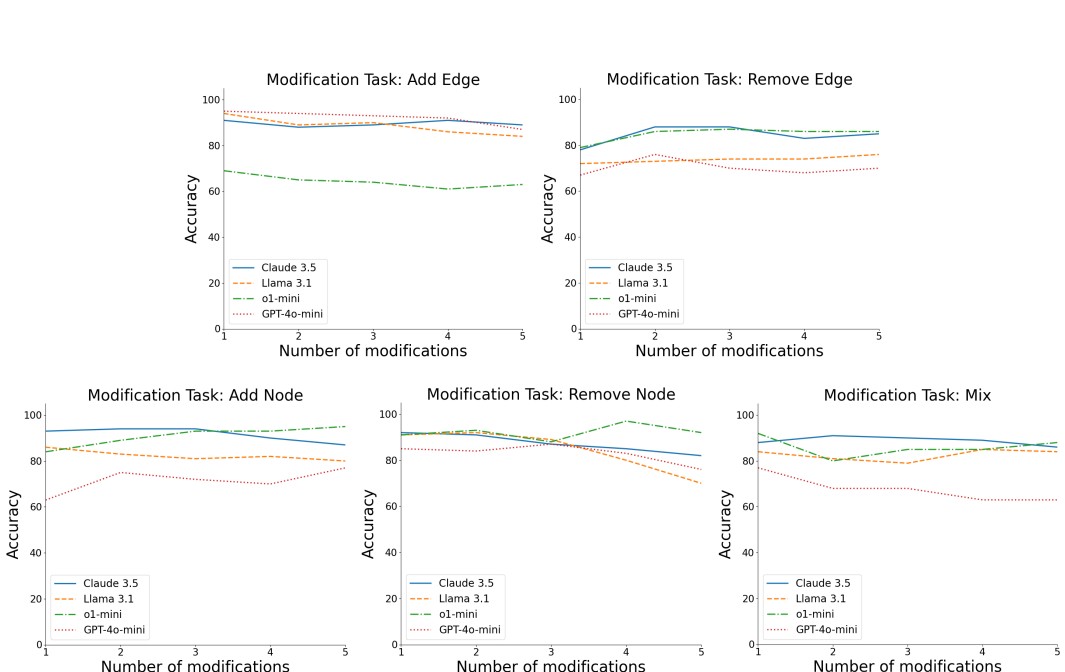

Figure 23: Print Graph, Coauthorship.

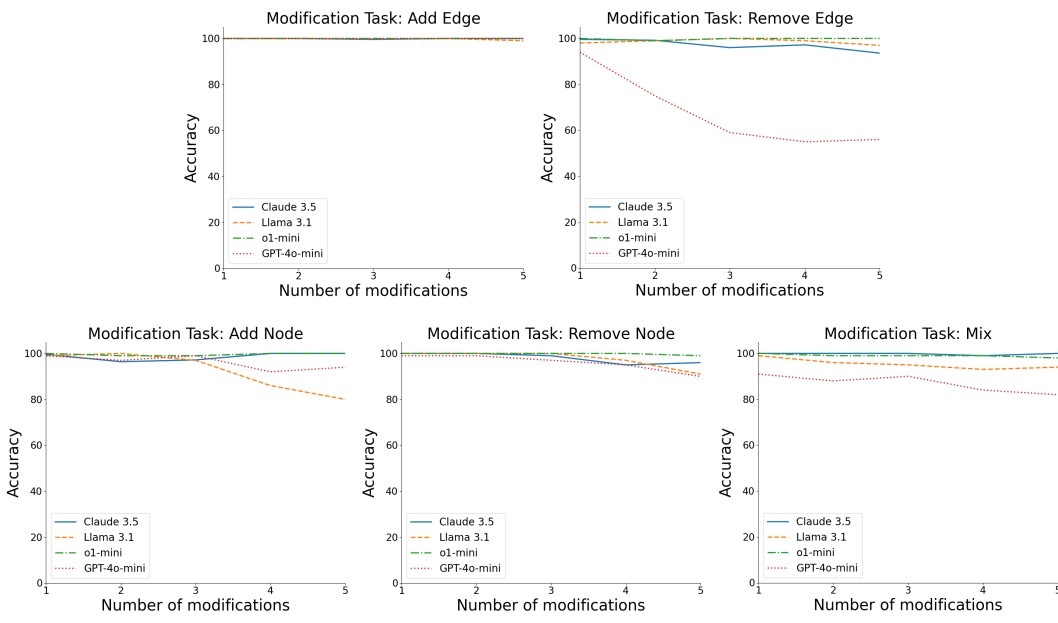

Figure 24: Print Graph, Incident List .

## A.7 EDGE DENSITY AND GRAPH SIZE ABLATION

In this section, we investigate how varying edge density and graph size impact model performance. For this analysis, we evaluate graphs with sizes $n \in \{7, 10, 15, 20\}$ and edge densities $p \in \{0.1, 0.5, 0.9\}$. For each combination of size and density, we generate 100 unique input graphs and follow the same procedure outlined in Algorithm 6 to create additional examples, focusing specifically on the **Print Graph** task. All evaluations are conducted using Claude 3.5 Sonnet with the **Adjacency Matrix** encoder.

With the **Add Edge** modification (Figure 25), the model maintains strong performance, with slight drops in performance observed as the number of nodes in the graph increases. Interestingly, the model performs poorly when asked to add a single edge for low-density graphs, indicating that the sparsity of the matrix may be influencing the model's ability to update the correct 0 entry, an issue that the model seems to correct as it makes more modifications.

The **Remove Edge** modification (Figure 26) shows strong overall performance, but an inverse trend compared to **Add Edge** is observed. As the number of nodes increases, the model struggles at removing edges from high-density graphs, and this challenge becomes more pronounced with an increasing number of modifications. This suggests that the model struggles to accurately identify the correct 1 entry to update in the adjacency matrix for dense graphs.

The **Add Node** modification (Figure 27) demonstrates very strong performance initially, but accuracy declines as the number of nodes in the graph increases. At moderate edge densities on large graphs ($n = 20$), the model performs well, but its performance begins to falter at higher densities. For low-density graphs, the model struggles more significantly, with accuracy further decreasing as the number of modifications increases.

Finally, for the **Remove Node** modification (Figure 28), both graph density and size significantly impact performance. The model struggles increasingly as the number of nodes grows. Interestingly, performance is lowest for low-density graphs, while high-density graphs tend to yield the best results overall across all graph sizes.

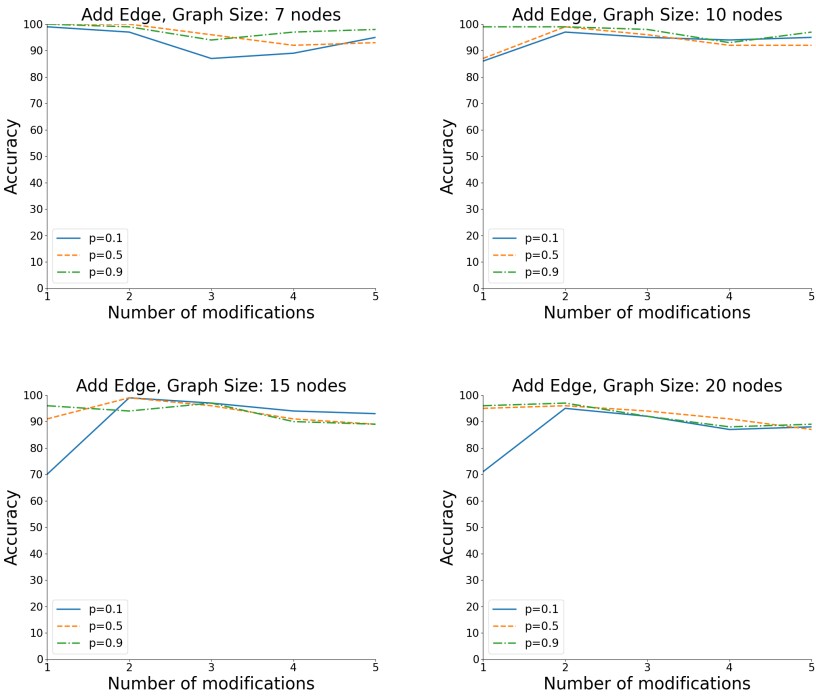

Figure 25: Add Edge Ablation.

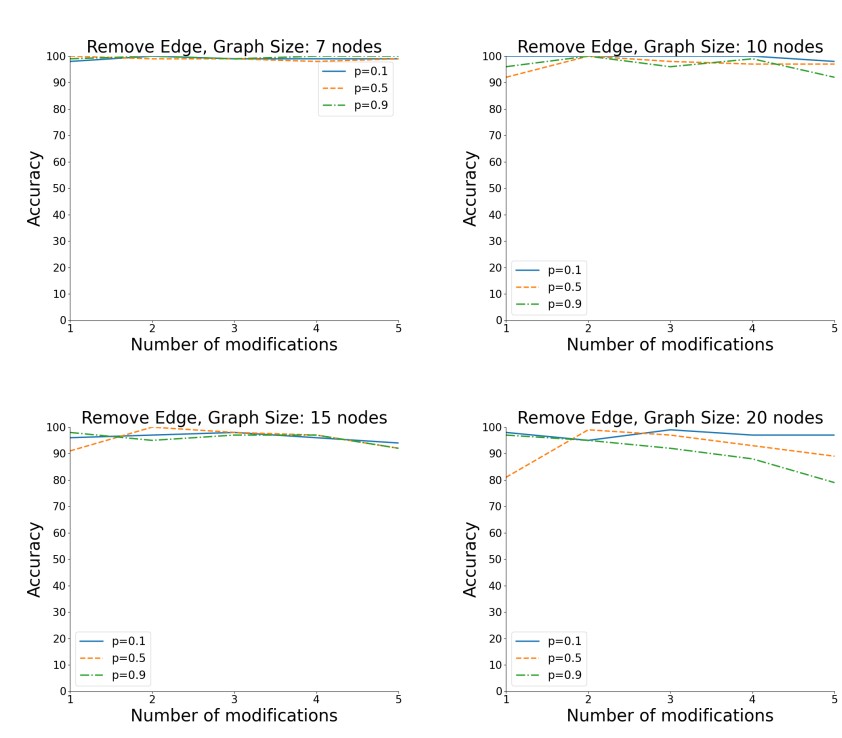

Figure 26: Remove Edge Ablation.

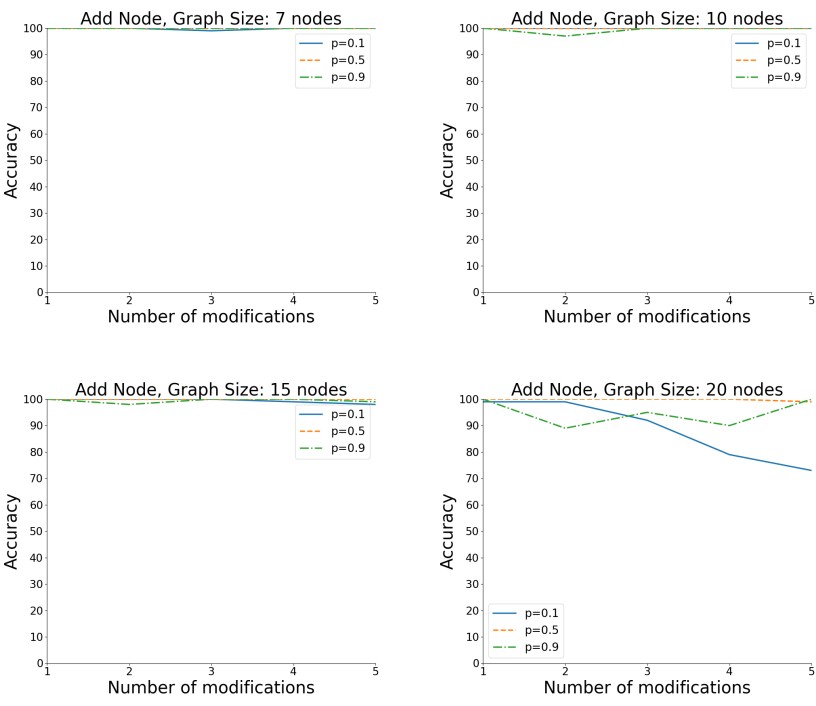

Figure 27: Add Node Ablation.

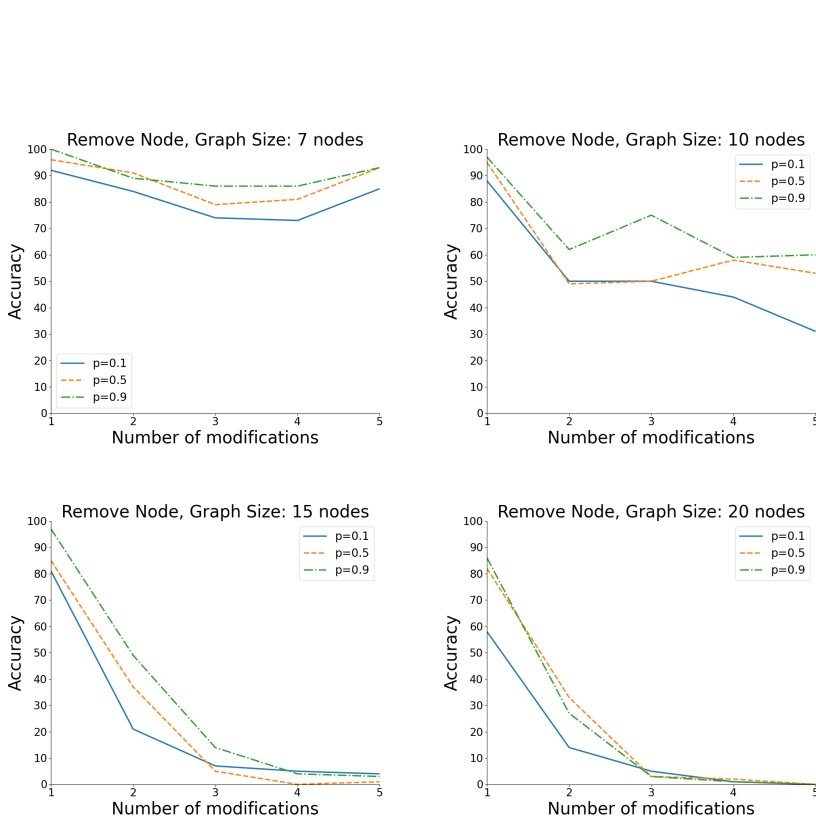

Figure 28: Remove Node Ablation.

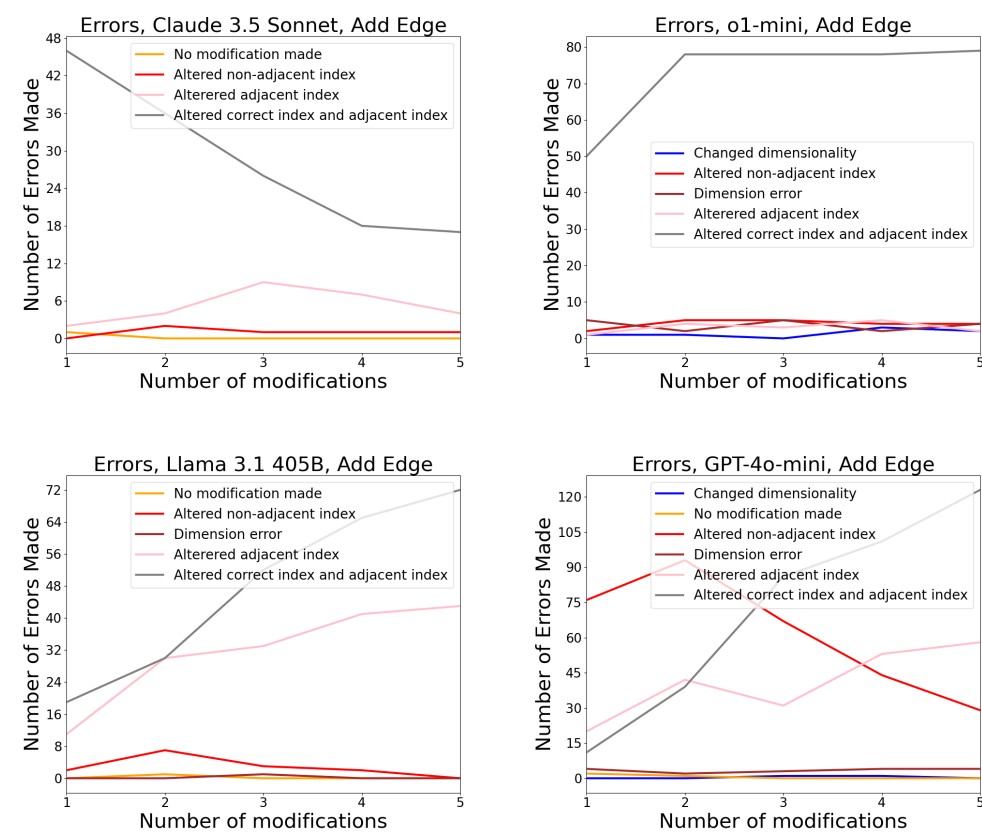

Figure 29: Error Types on Add Edge Modification.

## A.8 ERROR ANALYSIS

In this section, we analyze the types and frequencies of errors made by the four benchmark models on the **Print Graph** task using the **Adjacency Matrix** encoder. These error types help highlight each model's performance and challenges in graph modification tasks.

### A.8.1 ADD EDGE

Figure 29 shows the different types of errors all four models make on the **Add Edge** modification. We observe the following error types:

- **Altered correct index and adjacent index**: This error occurs when the model correctly identifies the indices to modify in the adjacency matrix but also erroneously adds an edge to at least one adjacent index. This is the most frequent error type across all models. Both Llama 3.1 405B and GPT-4o mini exhibit an increase in this error type as the number of modifications grows, indicating a scaling issue. For both o1-mini and Claude 3.5 Sonnet, this error overwhelmingly dominates their performance, as they both make few other types of errors. Interestingly, Claude 3.5 Sonnet reduces this error frequency as the number of modifications increases. This reduction may explain the model's improved performance under higher problem complexity observed in Figure 2, as it hallucinates fewer erroneous adjacent edges.

- **Altered adjacent index**: In this case, the model modifies an adjacent index without altering the correct one. This error becomes more common for Llama 3.1 405B and GPT-4o mini with an increasing number of modifications. Claude 3.5 Sonnet and o1-mini both maintain relatively constant and lower rates of this error.

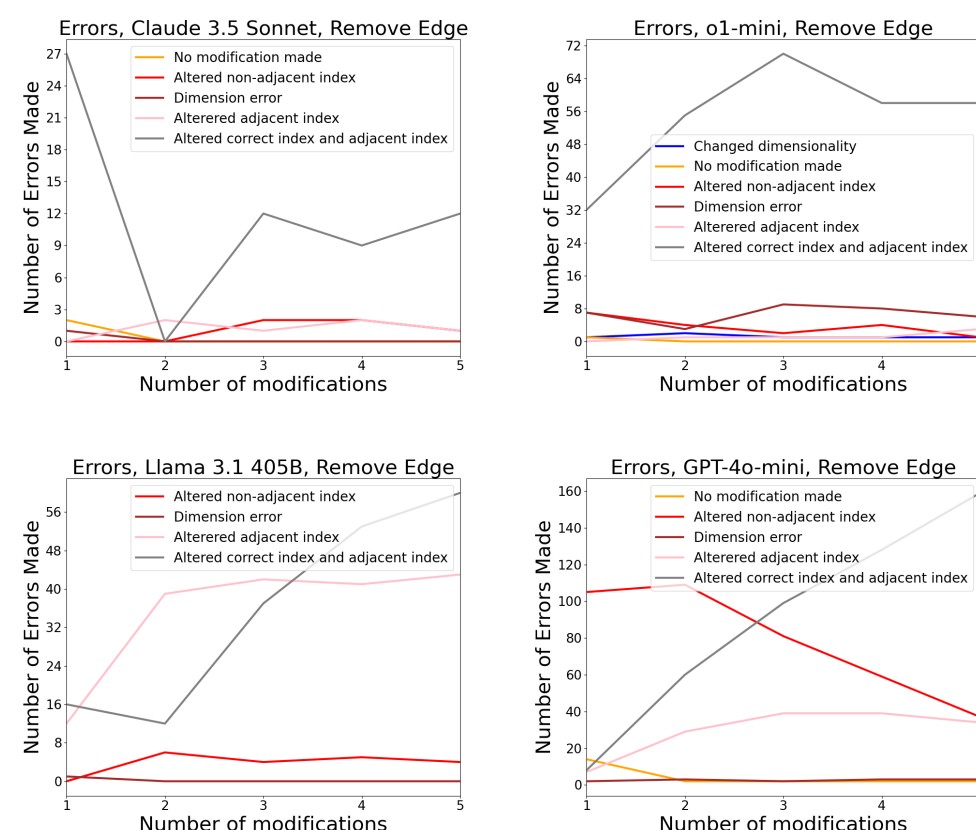

Figure 30: Error Types on Remove Edge Modification.

- **Altered non-adjacent index**: This error involves modifying an index that is not adjacent to the correct index. While rare in most models, it is more prominent in GPT-4o mini, suggesting that this error decreases with larger model sizes and improved reasoning capabilities. Interestingly, GPT-4o mini makes this error less often as the number of modifications increases. As shown by Figure 2, GPT-4o mini's performance on the **Add Edge** modification still decreases across the number of modifications, suggesting that the model's edits become increasingly closer to the correct indices as the complexity of the problem increases.

- **No modification made**: This occurs when the model outputs the unmodified input adjacency matrix. It is rare across all models and entirely absent in o1-mini.

- **Dimension error**: This error arises when the model returns an object that is not a valid matrix, and in our analysis this object mostly takes the form of a list of rows with inconsistent column counts. While infrequent, this error is never produced by Claude 3.5 Sonnet.

- **Changed dimensionality**: Here, the model outputs a well-defined matrix but with incorrect dimensions. This error occurs only occasionally in o1-mini and GPT-4o mini.

A.8.2 REMOVE EDGE

Figure 30 illustrates the types and frequencies of errors made by the models on the Remove Edge modification, demonstrating a similar error distribution to the Add Edge modification:

- **Altered correct index and adjacent index**: This remains the most common error across models. Both Llama 3.1 405B and GPT-4o mini exhibit an increase in this error as the number of modifications grows, reflecting a recurring challenge with hallucinating adjacent edges. For o1-mini and Claude 3.5 Sonnet, this error type also dominates.

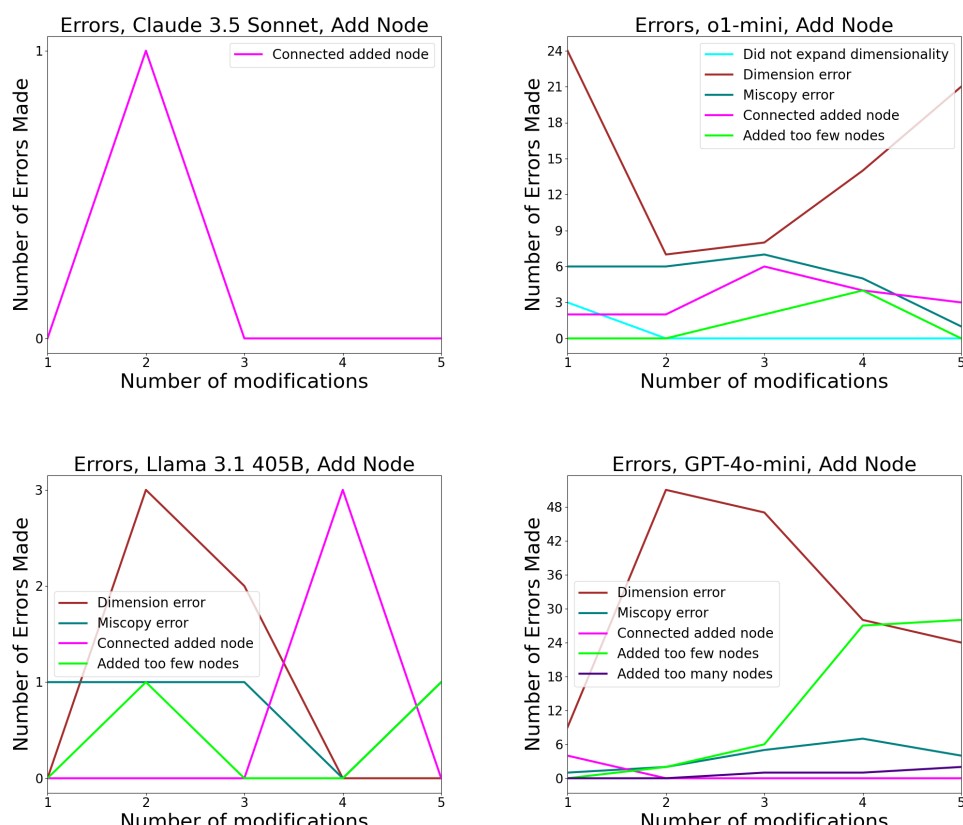

Figure 31: Error Types on Add Node Modification.

- **Altered adjacent index**: Similar to the Add Edge modification, this error is most prominent in Llama 3.1 405B and GPT-4o mini, while remaining rare for o1-mini and Claude 3.5 Sonnet.

- **Altered non-adjacent index**: Again, rare in most models, and most prominent in GPT-4o mini, but again we observe that the frequency of this error decreases as number of mods increases. As in the Add Edge modification, this error is rare across most models but is most frequently observed in GPT-4o mini. Again, we observe that GPT-4o mini makes this error less often as the number of modifications increase.

- **No modification made**: Consistent with previous observations, this error is rare across all model, with the only difference being that now Llama 3.1 405B never makes this error as opposed to o1-mini.

- **Dimension error**: We again observe that this error is rarely made across all models.

- **Changed dimensionality**: This error is absent across all models except o1-mini, which rarely makes this error.

### A.8.3 ADD NODE

Figure 31 showcases the types and frequencies of errors made by the models during the Add Node modification. The plots highlight the strong performance of Claude 3.5 Sonnet and Llama 3.1 405B, which make very few errors overall:

- **Connected added node**: This error involves incorrectly connecting the newly added node to at least one existing node. It is rare across all models, with Claude 3.5 Sonnet making this error only once. o1-mini produces this error slightly more frequently than other models.

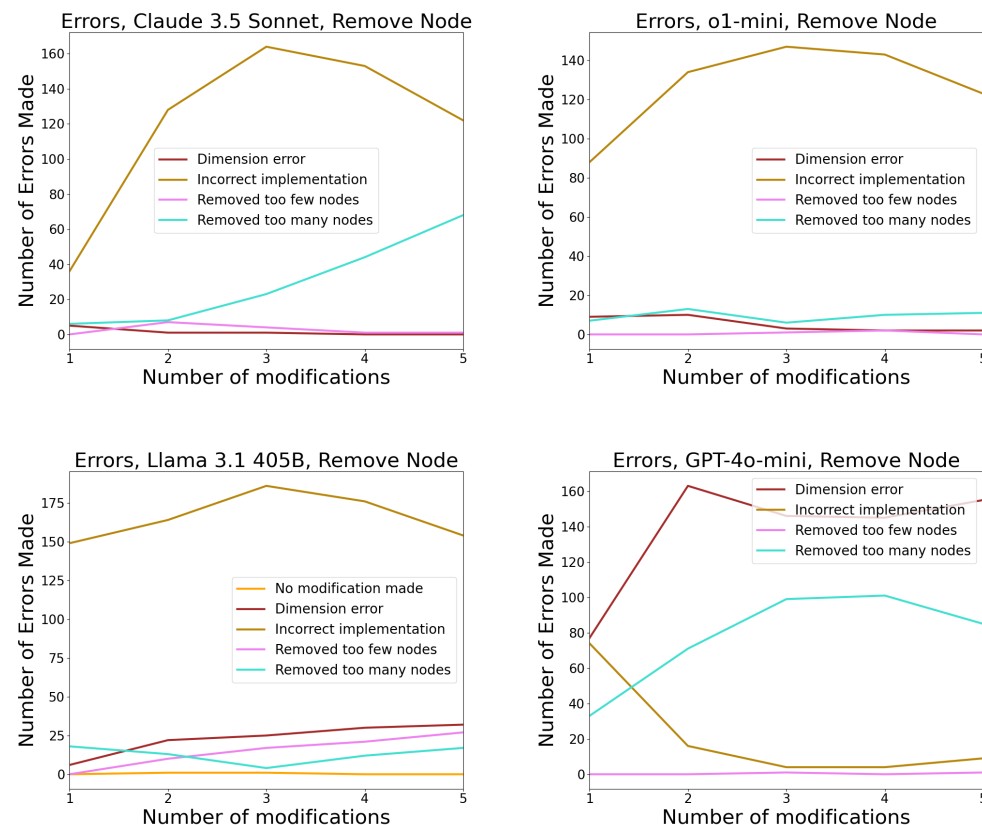

Figure 32: Error Types on Remove Node Modification.

- **Miscopy error**: This error occurs when the model correctly adds a new row and column of zeros to the adjacency matrix but mistakenly modifies at least one existing edge. It is an uncommon error, though o1-mini again displays a slightly higher error rate compared to others.

- **Added too few nodes**: When the required number of modifications is $k$, this error arises when the model adds fewer than $k$ nodes. This error is rare but becomes more prevalent for GPT-4o mini as $k$ increases, indicating that the model struggles with accurately tracking the number of modifications needed as the task complexity grows, potentially due to challenges in state management.

- **Added too many nodes**: This error occurs when the model adds more than the specified $k$ nodes to the graph. It is an infrequent error type, observed only in GPT-4o mini.

- **Dimension error**: This error is a frequent issue for both o1-mini and GPT-4o mini, with GPT-4o mini making this error more often, yet not as often at both high and low values of $k$.

### A.8.4 REMOVE NODE

Figure 32 illustrates the types of errors encountered in the Remove Node modification, the most challenging modification in GraphModQA:

- **Removed too many nodes**: This error arises when the model removes more than the required $k$ nodes. It is less frequent in o1-mini and Llama 3.1 405B but occurs at a high frequency in GPT-4o mini and Claude 3.5 Sonnet, with Claude 3.5 Sonnet exhibiting an increase in this error as $k$ grows.

- **Removed too few nodes**: This error occurs when the model removes fewer than $k$ nodes. It is generally infrequent, though Llama 3.1 405B makes this error slightly more often than the other models.

- **No modification made**: Only Llama 3.1 405B produces this error, and produces it very rarely.

- **Dimension error**: This error is made by Claude 3.5 Sonnet and o1-mini, while Llama 3.1 405B produces it slightly more often. However, this is the most frequent error for GPT-4o mini, indicating that it struggles significantly with maintaining a valid matrix structure and returning a mathematically well-defined object.

- **Incorrect implementation**: This error occurs when the model removes the correct number of rows and columns, but the internal connections of the resulting matrix deviate substantially from the solution matrix. This is the most common error type across all models except GPT-4o mini, which struggles to produce valid outputs at all due to frequent dimension errors. o1-mini makes this error less frequently than Claude 3.5 Sonnet and Llama 3.1 405B, contributing to its superior overall performance on the Remove Node modification as seen in Figure 2.

A.9 ILLUSTRATING PROMPTS AND MODEL OUTPUTS

In this section, we show example input prompts and model outputs from the three in-context methods we evaluate on GraphModQA. We evaluated Claude 3.5 Sonnet on a single example using the **Adjacency Matrix** encoding, involving three **Add Edge** modifications followed by a **Print Graph** task. Figures 33 and 34 show the input prompt and model output respectively using zero-shot prompting, Figures 35 and 36 show the input prompt and model output respectively using MAP prompting, and Figures 37 and 38 show the input prompt and model output respectively using CoT prompting with one example.

```
Zero-shot Prompt

The following matrix represents the adjacency matrix of an
    undirected graph, where the first row corresponds to node 0, the
    second row corresponds to node 1, and so on:

[[0 1 0 0 0 0 0 0 0 1 0 0 0 0 0]
 [1 0 1 1 0 0 1 0 1 0 1 0 0 0 0]
 [0 1 0 0 0 0 0 0 0 0 1 0 1 0 0]
 [0 1 0 0 0 0 0 0 0 0 0 0 1 0 0]
 [0 0 0 0 0 1 0 0 0 0 0 0 0 1 0]
 [0 0 0 0 1 0 0 0 0 1 0 0 0 0 1]
 [0 1 0 0 0 0 0 0 1 0 0 1 0 0 0]
 [0 0 0 0 0 0 0 0 0 0 0 0 1 1 1]
 [0 1 0 0 0 0 1 0 0 1 0 0 0 0 0]
 [1 0 0 0 0 1 0 0 1 0 0 0 0 0 1]
 [0 1 1 0 0 0 0 0 0 0 0 0 0 0 0]
 [0 0 0 0 0 0 1 0 0 0 0 0 0 0 1]
 [0 0 1 1 0 0 0 1 0 0 0 0 0 0 0]
 [0 0 0 0 1 0 0 1 0 0 0 0 0 0 0]
 [0 0 0 0 0 1 0 1 0 1 0 1 0 0 0]]

Perform the following operations on the graph:
1: Add an edge between node 14 and node 8.
2: Add an edge between node 8 and node 3 in the resulting graph of
    operation 1.
3: Add an edge between node 2 and node 8 in the resulting graph of
    operation 2.
Q: What is the final resulting adjacency matrix?
A:
```

Figure 33: An example zero-shot prompt. The prompt consists of three **Add Edge** modification, followed by the **Print Graph** task using the **Adjacency Matrix** encoding.

LLM Output on Zero-shot Prompt

```
To answer this question, I'll start with the given adjacency matrix
    and perform the requested operations one by one. After each
    operation, I'll update the matrix. Here's the process:

1. Add an edge between node 14 and node 8:
   This adds a 1 in position (14,8) and (8,14) of the matrix.

2. Add an edge between node 8 and node 3:
   This adds a 1 in position (8,3) and (3,8) of the matrix.

3. Add an edge between node 2 and node 8:
   This adds a 1 in position (2,8) and (8,2) of the matrix.

After performing these operations, the final resulting adjacency
matrix is:

[[0 1 0 0 0 0 0 0 0 1 0 0 0 0 0]
 [1 0 1 1 0 0 1 0 1 0 1 0 0 0 0]
 [0 1 0 0 0 0 0 1 0 0 1 0 1 0 0]
 [0 1 0 0 0 0 0 1 0 0 0 0 1 0 0]
 [0 0 0 0 0 1 0 0 0 0 0 0 0 1 0]
 [0 0 0 0 1 0 0 0 0 1 0 0 0 0 1]
 [0 1 0 0 0 0 0 0 1 0 0 1 0 0 0]
 [0 0 0 0 0 0 0 0 0 0 0 0 1 1 1]
 [0 1 1 1 0 0 1 0 0 1 0 0 0 0 1]
 [1 0 0 0 0 1 0 0 1 0 0 0 0 0 1]
 [0 1 1 0 0 0 0 0 0 0 0 0 0 0 0]
 [0 0 0 0 0 0 1 0 0 0 0 0 0 0 1]
 [0 0 1 1 0 0 0 1 0 0 0 0 0 0 0]
 [0 0 0 0 1 0 0 1 0 0 0 0 0 0 0]
 [0 0 0 0 0 1 0 1 1 1 0 1 0 0 0]]

This matrix represents the final state of the graph after all the
requested operations have been performed.
```

**Incorrect!**

Figure 34: The output of Claude 3.5 Sonnet on the example prompt from Figure 33. Here, the model got the question wrong, as the model modified positions (2, 7) and (3, 7) instead of positions (2, 8) and (3, 8), despite modifying positions (8, 2) and (8, 3) correctly. Correctly modified indicies are highlighted in green while incorrectly modified indicies are highlighted in red.

---

**MAP Prompt**

```
The following matrix represents the adjacency matrix of an
    undirected graph, where the first row corresponds to node 0, the
    second row corresponds to node 1, and so on:

[[0 1 0 0 0 0 0 0 0 1 0 0 0 0 0]
 [1 0 1 1 0 0 1 0 1 0 1 0 0 0 0]
 [0 1 0 0 0 0 0 0 0 0 0 1 0 1 0 0]
 [0 1 0 0 0 0 0 0 0 0 0 0 0 1 0 0]
 [0 0 0 0 0 1 0 0 0 0 0 0 0 0 1 0]
 [0 0 0 0 1 0 0 0 0 1 0 0 0 0 1]
 [0 1 0 0 0 0 0 0 1 0 0 1 0 0 0]
 [0 0 0 0 0 0 0 0 0 0 0 0 0 1 1 1]
 [0 1 0 0 0 0 1 0 0 1 0 0 0 0 0]
 [1 0 0 0 0 1 0 0 1 0 0 0 0 0 1]
 [0 1 1 0 0 0 0 0 0 0 0 0 0 0 0]
 [0 0 0 0 0 0 1 0 0 0 0 0 0 0 1]
 [0 0 1 1 0 0 0 1 0 0 0 0 0 0 0]
 [0 0 0 0 1 0 0 1 0 0 0 0 0 0 0]
 [0 0 0 0 0 1 0 1 0 1 0 1 0 0 0]]

Perform the following operations on the graph:
1: Add an edge between node 14 and node 8.
2: Add an edge between node 8 and node 3 in the resulting graph of
    operation 1.
3: Add an edge between node 2 and node 8 in the resulting graph of
    operation 2.
Q: What is the final resulting adjacency matrix? For each operation
    , write out the entire resulting adjacency matrix.
A:
```

Figure 35: An example MAP prompt, created by appending the phrase *For each operation, write out the entire resulting adjacency matrix.* to the prompt from Figure 33.

```
LLM Output on MAP Prompt

I'll perform the operations step by step and provide the resulting
    adjacency matrix after each operation.

Operation 1: Add an edge between node 14 and node 8

[[0 1 0 0 0 0 0 0 0 0 1 0 0 0 0 0]
[1 0 1 1 0 0 1 0 1 0 1 0 0 0 0 0]
...
[0 0 0 0 1 0 0 1 0 0 0 0 0 0 0 0]
[0 0 0 0 0 1 0 1 1 1 0 1 0 0 0 0]]

Operation 2: Add an edge between node 8 and node 3

[[0 1 0 0 0 0 0 0 0 0 1 0 0 0 0 0]
 [1 0 1 1 0 0 1 0 1 0 1 0 0 0 0 0]
 [0 1 0 0 0 0 0 0 0 0 1 0 1 0 0]
 [0 1 0 0 0 0 0 0 1 0 0 0 1 0 0]
 [0 0 0 0 0 1 0 0 0 0 0 0 0 1 0]
 [0 0 0 0 1 0 0 0 0 1 0 0 0 0 1]
 [0 1 0 0 0 0 0 0 1 0 0 1 0 0 0]
 [0 0 0 0 0 0 0 0 0 0 0 0 1 1 1]
 [0 1 0 1 0 0 1 0 0 1 0 0 0 0 1]
 [1 0 0 0 0 1 0 0 1 0 0 0 0 0 1]
 [0 1 1 0 0 0 0 0 0 0 0 0 0 0 0]
 [0 0 0 0 0 0 1 0 0 0 0 0 0 0 1]
 [0 0 1 1 0 0 0 1 0 0 0 0 0 0 0]
 [0 0 0 0 1 0 0 1 0 0 0 0 0 0 0]
 [0 0 0 0 0 1 0 1 1 1 0 1 0 0 0]]

Operation 3: Add an edge between node 2 and node 8

Here's the final resulting adjacency matrix:

[[0 1 0 0 0 0 0 0 0 0 1 0 0 0 0 0]
 [1 0 1 1 0 0 1 0 1 0 1 0 0 0 0 0]
 [0 1 0 0 0 0 0 0 1 0 1 0 1 0 0]
 [0 1 0 0 0 0 0 0 1 0 0 0 1 0 0]
 [0 0 0 0 0 1 0 0 0 0 0 0 0 1 0]
 [0 0 0 0 1 0 0 0 0 1 0 0 0 0 1]
 [0 1 0 0 0 0 0 0 1 0 0 1 0 0 0]
 [0 0 0 0 0 0 0 0 0 0 0 0 1 1 1]
 [0 1 1 1 0 0 1 0 0 1 0 0 0 0 1]
 [1 0 0 0 0 1 0 0 1 0 0 0 0 0 1]
 [0 1 1 0 0 0 0 0 0 0 0 0 0 0 0]
 [0 0 0 0 0 0 1 0 0 0 0 0 0 0 1]
 [0 0 1 1 0 0 0 1 0 0 0 0 0 0 0]
 [0 0 0 0 1 0 0 1 0 0 0 0 0 0 0]
 [0 0 0 0 0 1 0 1 1 1 0 1 0 0 0]]

Correct!
```

Figure 36: The output of Claude 3.5 Sonnet on the example MAP prompt from Figure 35. The first intermediate matrix is truncated in order to fit the entire output on the page. Here, the model got the question right, and the correctly modified indicies are highlighted in green in the final matrix.

```
CoT Prompt with one example

The following matrix represents the adjacency matrix of an
    undirected graph, where the first row corresponds to node 0, the
     second row corresponds to node 1, and so on:
[[0 1 1 0 0 0 1 0 1 0 1 1 0 0 0 0]
 [1 0 1 0 0 1 0 0 1 0 0 1 0 1 0 0]
 [1 1 0 1 1 0 1 0 0 0 0 0 0 0 0 1]
 [0 0 1 0 0 1 1 0 1 0 0 0 0 0 0 0]
 [0 0 1 0 0 0 0 0 0 1 0 0 1 1 0 1]
 [0 1 0 1 0 0 0 0 1 0 0 0 0 1 1 0]
 [1 0 1 1 0 0 0 0 0 0 1 0 0 1 0 1]
 [0 0 0 0 0 0 0 0 0 0 0 0 0 1 0 1 0]
 [1 1 0 1 0 1 0 0 0 0 1 1 0 0 0 0]
 [0 0 0 0 1 0 0 0 1 0 0 0 0 1 0 0]
 [1 0 0 0 0 0 1 0 1 0 0 0 0 1 1 0]
 [1 1 0 0 0 0 0 0 0 0 0 0 0 1 1 0 0]
 [0 0 0 0 1 0 0 1 0 0 0 1 0 0 1 0]
 [0 1 0 0 1 1 1 1 0 0 1 1 1 0 0 0 0]
 [0 0 0 0 0 0 1 0 1 0 0 1 0 1 0 0 0]
 [0 0 1 0 1 0 1 0 0 0 0 0 0 0 0 0]]
Perform the following operations on the graph:
1: Add an edge between node 15 and node 13.
2: Add an edge between node 15 and node 3 in the resulting graph of
    operation 1.
3: Add an edge between node 10 and node 7 in the resulting graph of
    operation 2.
Q: What is the final resulting adjacency matrix? Write out the
    entire final resulting adjacency matrix.
A: After changing both entries [15, 13] and [13, 15] from 0 to 1,
    both entries [15, 3] and [3, 15] from 0 to 1, and both entries
    [10, 7] and [7, 10] from 0 to 1, the resulting adjacency matrix
    is as defined below:
[[0 1 1 0 0 0 1 0 1 0 1 1 0 0 0 0]
 [1 0 1 0 0 1 0 0 1 0 0 1 0 1 0 0]
 [1 1 0 1 1 0 1 0 0 0 0 0 0 0 0 1]
 [0 0 1 0 0 1 1 0 1 0 0 0 0 0 0 1]
 [0 0 1 0 0 0 0 0 0 1 0 0 1 1 0 1]
 [0 1 0 1 0 0 0 0 1 0 0 0 0 1 1 0]
 [1 0 1 1 0 0 0 0 0 0 0 1 0 0 1 0 1]
 [0 0 0 0 0 0 0 0 0 0 1 0 1 0 1 0]
 [1 1 0 1 0 1 0 0 0 0 1 1 0 0 0 0]
 [0 0 0 0 1 0 0 0 1 0 0 0 0 1 0 0]
 [1 0 0 0 0 0 1 1 1 0 0 0 0 1 1 0]
 [1 1 0 0 0 0 0 0 0 0 0 0 0 1 1 0 0]
 [0 0 0 0 1 0 0 1 0 0 0 1 0 0 1 0]
 [0 1 0 0 1 1 1 1 0 0 1 1 1 0 0 0 1]
 [0 0 0 0 0 0 1 0 1 0 0 1 0 1 0 0 0]
 [0 0 1 1 1 0 1 0 0 0 0 0 0 1 0 0]]

The following matrix represents the adjacency matrix of an
    undirected graph, where the first row corresponds to node 0, the
     second row corresponds to node 1, and so on:
...
Q: What is the final resulting adjacency matrix?
A:
```

Figure 37: An example CoT prompt with one example, created by adding the example before the
prompt (truncated) from Figure 33.

LLM Output on CoT Prompt

```
To solve this problem, we'll perform the operations one by one and
    update the adjacency matrix accordingly. Remember that for an
    undirected graph, when we add an edge between two nodes, we need
     to update both corresponding entries in the matrix.

Let's perform the operations:

1. Add an edge between node 14 and node 8:
   Update [14][8] and [8][14] from 0 to 1

2. Add an edge between node 8 and node 3:
   Update [8][3] and [3][8] from 0 to 1

3. Add an edge between node 2 and node 8:
   Update [2][8] and [8][2] from 0 to 1

After applying these changes, the final resulting adjacency matrix
    is:

[[0 1 0 0 0 0 0 0 0 1 0 0 0 0 0]
 [1 0 1 1 0 0 1 0 1 0 1 0 0 0 0]
 [0 1 0 0 0 0 0 0 1 0 1 0 1 0 0]
 [0 1 0 0 0 0 0 0 1 0 0 0 1 0 0]
 [0 0 0 0 0 1 0 0 0 0 0 0 0 1 0]
 [0 0 0 0 1 0 0 0 0 1 0 0 0 0 1]
 [0 1 0 0 0 0 0 0 1 0 0 1 0 0 0]
 [0 0 0 0 0 0 0 0 0 0 0 0 1 1 1]
 [0 1 1 1 0 0 1 0 0 1 0 0 0 0 1]
 [1 0 0 0 0 1 0 0 1 0 0 0 0 0 1]
 [0 1 1 0 0 0 0 0 0 0 0 0 0 0 0]
 [0 0 0 0 0 0 1 0 0 0 0 0 0 0 1]
 [0 0 1 1 0 0 0 1 0 0 0 0 0 0 0]
 [0 0 0 0 1 0 0 1 0 0 0 0 0 0 0]
 [0 0 0 0 0 1 0 1 1 1 0 1 0 0 0]]

This is the final resulting adjacency matrix after performing all
    the specified operations.
```

**Correct!**

Figure 38: The output of Claude 3.5 Sonnet on the CoT prompt from Figure 37. Here, the model got the question right, and the correctly modified indicies are highlighted in green in the final matrix.

Table 1: Accuracy percentages (mean ± s.d) on graph property questions from Fatemi et al. (2023).

| Graph Encoder | Final Task | GPT-4o mini | Llama 3.1 | Claude 3.5 | o1-mini |
|---|---|---|---|---|---|
| Adjacency List | Node Count | 86.6 | 89.8 | 95.4 | 99.8 |
| | Edge Count | 30.4 | 48.8 | 54.8 | 93.6 |
| | Node Degree | 95.2 | 100.0 | 100.0 | 98.4 |
| | Edge Existence | 71.8 | 70.6 | 86.8 | 66.0 |
| | Connected Nodes | 97.8 | 100.0 | 100.0 | 98.2 |
| | Cycle | 90.4 | 91.0 | 95.0 | 99.0 |
| | **Average** | **78.7 ± 23.2** | **83.4 ± 18.3** | **88.7 ± 15.8** | **92.5 ± 12.02** |
| Incident | Node Count | 100.0 | 99.8 | 100.0 | 100.0 |
| | Edge Count | 30.0 | 60.4 | 76.2 | 99.0 |
| | Node Degree | 99.2 | 99.2 | 100.0 | 99.6 |
| | Edge Existence | 95.2 | 91.0 | 99.8 | 66.6 |
| | Connected Nodes | 99.8 | 100.0 | 100.0 | 100.0 |
| | Cycle | 86.2 | 87.4 | 88.4 | 98.8 |
| | **Average** | **85.1 ± 25.1** | **91.3 ± 14.2** | **94.1 ± 9.0** | **94.0 ± 12.3** |
| Friendship | Node Count | 99.6 | 98.8 | 100.0 | 100.0 |
| | Edge Count | 27.6 | 49.2 | 57.0 | 86.8 |
| | Node Degree | 91.6 | 98.2 | 100.0 | 98.0 |
| | Edge Existence | 73.0 | 76.0 | 77.4 | 66.0 |
| | Connected Nodes | 87.8 | 93.4 | 95.2 | 92.6 |
| | Cycle | 91.6 | 91.8 | 95.6 | 99.8 |
| | **Average** | **78.5 ± 24.1** | **84.6 ± 17.5** | **87.5 ± 15.7** | **90.5 ± 11.9** |
| Coauthorship | Node Count | 99.0 | 99.0 | 95.6 | 100.0 |
| | Edge Count | 27.4 | 42.8 | 54.2 | 78.2 |
| | Node Degree | 88.0 | 94.0 | 99.6 | 96.4 |
| | Edge Existence | 85.6 | 84.2 | 88.6 | 65.0 |
| | Connected Nodes | 75.2 | 91.6 | 98.2 | 93.4 |
| | Cycle | 92.4 | 95.6 | 100.0 | 99.4 |
| | **Average** | **77.9±23.7** | **84.5±19.2** | **89.4±16.2** | **88.7 ± 12.9** |
| Expert | Node Count | 87.4 | 82.8 | 79.2 | 99.4 |
| | Edge Count | 35.2 | 52.2 | 62.8 | 95.0 |
| | Node Degree | 95.8 | 99.8 | 100.0 | 99.4 |
| | Edge Existence | 67.0 | 66.8 | 100.0 | 65.0 |
| | Connected Nodes | 97.4 | 97.4 | 95.2 | 89.4 |
| | Cycle | 86.2 | 85.8 | 96.0 | 98.0 |
| | **Average** | **78.2±21.6** | **80.8±16.7** | **88.9±13.6** | **91.0 ± 12.1** |
| Social Network | Node Count | 99.6 | 99.4 | 100.0 | 100.0 |
| | Edge Count | 26.4 | 48.0 | 57.8 | 81.8 |
| | Node Degree | 94.0 | 97.4 | 100.0 | 97.2 |
| | Edge Existence | 86.6 | 85.2 | 100.0 | 64.2 |
| | Connected Nodes | 85.4 | 92.8 | 94.8 | 93.4 |
| | Cycle | 91.8 | 90.4 | 93.6 | 98.6 |
| | **Average** | **80.6±24.7** | **85.5±17.4** | **91.0±15.1** | **89.2 ± 12.7** |
| Politician | Node Count | 99.4 | 100 | 99.6 | 100.0 |
| | Edge Count | 25.2 | 48.2 | 55.4 | 85.8 |
| | Node Degree | 94.0 | 97.0 | 99.8 | 98.6 |
| | Edge Existence | 88.8 | 81.6 | 71.0 | 66.0 |
| | Connected Nodes | 79.6 | 79.4 | 100.0 | 97.2 |
| | Cycle | 91.4 | 89.0 | 95.8 | 99.4 |
| | **Average** | **79.7±25.1** | **82.5±17.1** | **86.9±17.4** | **91.2 ± 12.2** |
| GoT | Node Count | 100.0 | 100.0 | 99.0 | 100.0 |
| | Edge Count | 26.8 | 46.0 | 57.4 | 84.8 |
| | Node Degree | 93.2 | 95.2 | 100.0 | 96.8 |
| | Edge Existence | 83.4 | 80.4 | 87.4 | 65.2 |
| | Connected Nodes | 68.4 | 95.8 | 100.0 | 94.6 |
| | Cycle | 91.4 | 95.6 | 94.8 | 100.0 |
| | **Average** | **77.2±24.6** | **85.5±18.7** | **89.8±15.1** | **90.2 ± 12.3** |
| SP | Node Count | 99.4 | 99.8 | 99.2 | 100.0 |
| | Edge Count | 26.0 | 44.4 | 59.2 | 86.0 |
| | Node Degree | 94.4 | 96.4 | 100.0 | 98.2 |
| | Edge Existence | 85.2 | 87.0 | 82.2 | 65.2 |
| | Connected Nodes | 74.2 | 98.6 | 100.0 | 98.0 |
| | Cycle | 91.4 | 93.0 | 95.0 | 99.6 |
| | **Average** | **78.4±24.8** | **86.5±19.3** | **89.3±14.8** | **91.2 ± 12.6** |

Table 2: Accuracy percentages (mean $\pm$ s.d) on graph property questions from Fatemi et al. (2023) for the adjacency matrix encoder. As this work was being conducted, the PaLM API was deprecated, and fortunately we were able to evaluate PaLM 2 L on the adjacency matrix encoder before this.

| Graph Encoder | Final Task | PaLM 2 L | GPT-4o mini | Llama 3.1 | Claude 3.5 | o1-mini |
|---|---|---|---|---|---|---|
| | Node Count | 55.4 | 98.4 | 100.0 | 100.0 | 98.4 |
| | Edge Count | 6.4 | 28.0 | 44.8 | 38.6 | 91.2 |
| | Node Degree | 28.6 | 73.4 | 88.6 | 98.6 | 99.2 |
| Adjacency Matrix | Edge Existence | 70.3 | 85.0 | 93.8 | 99.2 | 68.2 |
| | Connected Nodes | 8.4 | 84.8 | 98.2 | 99.0 | 98.8 |
| | Cycle | 49.6 | 87.8 | 87.6 | 92.8 | 100.0 |
| | **Average** | **36.5 $\pm$ 23.9** | **76.2 $\pm$ 22.8** | **85.5 $\pm$ 18.8** | **88.0 $\pm$ 22.2** | **92.6 $\pm$ 11.3** |

