# OpenReview forum: "Can Large Language Models Effectively Modify Graphs?"
_ICLR.cc/2025/Conference — Submitted to ICLR 2025_

### Official Review · Reviewer_2ZNt · 2024-10-26

**Soundness:** 2
**Presentation:** 2
**Contribution:** 2
**Rating:** 3
**Confidence:** 4

**Summary:**

This paper proposes a new benchmark dataset for graph understanding of LLMs. It contains QA pairs that first ask LLMs to modify graphs (e.g., add nodes or edges and remove nodes or edges), and then ask LLM questions regarding the structural properties of the graph (e.g., number of nodes). The authors also examine the performance of popular LLMs on this benchmark.

**Strengths:**

1. Using LLM to modify graph structure is a very interesting topic.
2. The proposed dataset is clearly introduced.

**Weaknesses:**

1. The contribution of the proposed dataset seems incremental. It uses the same graph property questions as in previous datasets but just has an additional step to ask the LLM to add or remove nodes and edges. However, why do we need LLMs to do specific modifications on graphs? Anyone can easily revise the adjacency matrix of a graph to add/remove nodes and edges (usually less than 10 lines of code ). Is there any real-world use case for such "modify and then analyze" graph questions?

2. Given the weak motivation of the proposed datasets, I think they are insufficient to examine the graph understanding ability of LLMs. A very interesting graph modification task for LLMs involves asking them to adjust a graph to achieve specific target properties. For example, using the minimum modifications to let a graph be disconnected or has at least K connected components, or using the minimum modifications to let a graph be isomorphism with another graph.

3. The authors mentioned many times "dynamic graph" in their paper, but it is still unclear to me which part of their datasets is related to understanding dynamic graphs. It seems the authors regard multi-step modification as a kind of "dynamic graph modification", but this is just related to modification rather than the "dynamic graph reasoning" claimed in the abstract.

4. Much important information is missing, such as the detailed analysis and statistics of the datasets, the prompt they used during experiments, and the hyper-parameter settings.

**Questions:**

See weakness

---

> ### Author Response · Authors · 2024-11-24
> **Comment by Authors**
>
> # Incremental Contribution
>
> We appreciate your points on the potentially incremental contributions of the paper and questioning the need for LLMs to modify graphs. Regarding your first point, the transition from static to dynamic graph algorithms is not a minor incremental step but represents a fundamental shift in approach. In algorithms research, dynamic graph problems constitute an entirely separate field due to their complexity and distinct challenges [1], demonstrating that converting a static problem into a dynamic one is not a minor incremental step. In fact, arguably the bigger worry with our paper is that we are only scratching the surface of exploring LLMs’ abilities of dynamic graph understanding, not in its scope being too incremental.
>
> Regarding your second point on whether or not we need LLMs to modify graphs, you’re of course correct in that code libraries provide efficient and reliable implementations for many graph tasks, let alone modification tasks, in a few lines of code. However, there is already a growing literature that looks at LLMs' intrinsic reasoning abilities on graphs (e.g., [2] and [3]), and as the literature explains, they are not advocating for replacing these established tools with large language models (LLMs) for practical graph tasks. Instead, they aim to evaluate LLMs’ reasoning capabilities within the context of graph tasks as a way to better understand and quantify their strengths and limitations in multi-step reasoning scenarios.
>
> We recognize a critical gap in the existing literature, where prior work has largely focused on basic static graph properties. Therefore, we add to this literature in a qualitatively new way by introducing the concept of graph modification as a more dynamic, multi-step reasoning challenge. This approach aligns with the broader goal of assessing and advancing LLMs’ ability to handle evolving, structured data representations, which is critical for dynamic domains such as social networks, where connections are frequently added or removed, and evolving knowledge bases, which must accommodate new or outdated information.
>
> # More Interesting Graph Modification Tasks
>
> We appreciate your suggestion of exploring more advanced graph modification tasks involving target property optimization (e.g., minimizing changes to disconnect a graph or achieving a specific number of connected components). While such tasks indeed represent intriguing extensions of graph modification, our focus in this work is to rigorously assess fundamental and interpretable capabilities in graph reasoning and dynamic graph manipulation. These foundational tasks are critical prerequisites for enabling LLMs to handle more complex scenarios such as target property optimization. We cannot reasonably evaluate more difficult scenarios when LLMs struggle to perform basic and atomic graph manipulations. Extending our methods to property-specific graph modifications, as suggested, is an exciting future direction, but it falls outside the scope of this work, which aims to establish a baseline for understanding and improving LLMs' abilities to perform graph modifications through foundational tasks.
>
> # More Important Information Needed
>
> We have added Section 4.3 in the new version of our paper, which details the total number of questions in GraphModQA (468,750 ), and how this number is derived.
>
> In Sections 4.1 and 5.2, we briefly mention how we sample the size and edge density of each graph. For readability, we consolidate this information, as well as other implementation details, in Section A.2 in the Appendix, which also includes our decoding temperature of zero, our only model hyper-parameter.
>
> Regarding the forms of the prompts we use, we have added Section A.9 which includes Figures 33 to 38. These figures illustrate examples of input prompts and model outputs corresponding to zero-shot prompting, CoT prompting, and MAP prompting.
>
> [1] Eppstein, David, Zvi Galil, and Giuseppe F. Italiano. "Dynamic graph algorithms." Algorithms and theory of computation handbook 1 (1999): 9-1.
>
> [2] Fatemi, Bahare, Jonathan Halcrow, and Bryan Perozzi. "Talk like a graph: Encoding graphs for large language models." arXiv preprint arXiv:2310.04560 (2023).
>
> [3] Wang, Heng, et al. "Can language models solve graph problems in natural language?." Advances in Neural Information Processing Systems 36 (2024).

---

> > ### Author Response · Authors · 2024-11-24
> > **Continuation of First Comment by Authors**
> >
> > # Dynamic Graph confusion
> >
> > Regarding your point on how our work is related to dynamic graphs, please firstly refer to the following paper [4]. Here is an excerpt from this paper that justifies our modification types:
> >
> >  *“(static) graphs and networks involve the following entities: V (a set of nodes), E (a set of edges), f (map vertices to numbers), and g (map edges to numbers). A dynamic graph is obtained when any of these four entities changes over time. Thus, there are four basic kinds of dynamic graphs.*
> >
> > - *In a node-dynamic graph or digraph, the set V varies with time. Thus, some nodes may be added or removed. When nodes are removed, the edges (or arcs) incident with them are also eliminated.*
> >
> > - *In an edge-dynamic (or arc-dynamic) graph or digraph, the set E varies with time. Thus, edges may be added or deleted from the graph or digraph.*
> >
> > - *In a node weighted dynamic graph, the function f varies with time; thus, the weights on the nodes also change.*
> >
> > - *In an edge or arc weighted dynamic graph or digraph, the function g varies with time.*
> >
> > - *In fully weighted dynamic graph, both functions f and g may vary with time.*
> >
> > *All combinations of the above types can occur.”*
> >
> > Therefore, in dynamic graphs, “nodes may be added or removed”, and “edges may be added or deleted”, which is why we chose these atomic graph modification operations.
> >
> > In addition, please refer to this Simons Institute talk by Monika Henzinger [5] that defines a “*dynamic graph algorithm*” as “*a data structure that maintains information about a graph while it is modified through local operations such as edge or vertex insertions or deletions.*” Therefore, graph modifications, such as adding or removing nodes and edges, are inherently related to dynamic graph reasoning, as they involve maintaining an accurate representation of the graph's evolving structure, and therefore our work directly addresses dynamic graph reasoning through systematic graph modifications.
> >
> > [4] Harary, Frank, and Gopal Gupta. "Dynamic graph models." Mathematical and Computer Modelling 25.7 (1997): 79-87.
> >
> > [5] https://simons.berkeley.edu/news/dynamic-graph-algorithms-what-we-know-what-we-dont

---

> ### Comment · Reviewer_2ZNt · 2024-11-26
> **Thanks for your response**
>
> The motivation of this dataset is mainly from the dynamic graphs. However, dynamic graphs in the real world have various evolution patterns and rules rather than randomly adding or deleting edges and nodes. It's unclear whether the modification queries of the proposed datasets are related to some real-world graph evolution properties. Moreover, I believe "dynamic graph reasoning" is more likely to predict the future graph structure rather than modify graph structure given human instructions. So the relationship between "graph modification" and "dynamic graph reasoning" is unconvincing to me. I will keep my score.

---

> ### Author Response · Authors · 2024-11-27
> **Responding to comment**
>
> Dear reviewer 2ZNt,
>
> We appreciate the response. It seems that there is some disagreement as to the definition of “dynamic graph reasoning”. We believe that we can clear this up by first defining what prior work defines as (static) “graph reasoning”, and then we can use this definition to define “dynamic graph reasoning”. The below explanation has been clarified in the newest version of the paper.
>
> Firstly, we would like to mention that prior work, namely [1] and [2], pose the question of if LLMs can reason on static graphs. The authors of these papers define "reasoning" in this sense as the ability of these LLMs to perform static graph algorithms (e.g connectivity, shortest path, etc.). A static graph algorithm takes in a static graph and outputs some property of the input graph. Therefore, the authors of [1] and [2] test the "graph reasoning" abilities of LLMs by evaluating if they are able to perform these static graph algorithms.
>
> From the talk we previously linked to above [3] and from extensive literature on the topic [4], a dynamic graph algorithm is composed of two parts:
> 1. “[A] data structure that maintains information about a graph while it is modified through local operations such as edge or vertex insertions or deletions.”
> 2. The ability to efficiently return properties of the updated graphs without recalculating from scratch after each modification.
>
> In our paper, we pose the question of if LLMs can reason on dynamic graphs. In order for an LLM to be able to reason on dynamic graphs, they must be able to do both of the above (accurately maintaining a data structure that undergoes a series of modifications, and efficiently return dynamic properties by leveraging precomputed information that is maintained during updates). As we show in our paper, LLMs struggle greatly on the first point, since they struggle with accurately maintaining one of the most commonly used data structures for representing graphs, the adjacency matrix, as the structure undergoes several key atomic modification operations. Therefore, graph modifications are inextricably linked with the ability to perform dynamic graph algorithms, and by staying consistent with definitions offered by prior work, the ability to perform dynamic graph algorithms can be similarly defined as "dynamic graph reasoning”.
>
> Understanding if LLMs can predict future graph structure and understand various evolutionary patterns found in dynamic graphs is certainly an interesting area of research, but we consider these problems to be out of scope, as we instead focus on the ability of LLMs to perform local operations.
>
> We hope that this clarifies the relationship between “graph modification” and “dynamic graph reasoning”.
>
> [1] Fatemi, Bahare, Jonathan Halcrow, and Bryan Perozzi. "Talk like a graph: Encoding graphs for large language models." arXiv preprint arXiv:2310.04560 (2023).
>
> [2] Wang, Heng, et al. "Can language models solve graph problems in natural language?." Advances in Neural Information Processing Systems 36 (2024).
>
> [3] https://simons.berkeley.edu/news/dynamic-graph-algorithms-what-we-know-what-we-dont
>
> [4] Eppstein, David, Zvi Galil, and Giuseppe F. Italiano. "Dynamic graph algorithms." Algorithms and theory of computation handbook 1 (1999): 9-1.

---

> > ### Author Response · Authors · 2024-12-01
> > **Invitation for Further Discussion**
> >
> > Dear Reviewer 2ZNt,
> >
> > As we come closer to the end of the discussion period, please let us know if you have any thoughts regarding our above comment addressing your concerns. We thank you very much for your efforts thus far.
> >
> > Authors

---

### Official Review · Reviewer_p6sQ · 2024-10-30

**Soundness:** 2
**Presentation:** 1
**Contribution:** 2
**Rating:** 3
**Confidence:** 4

**Summary:**

The paper proposes a graph modification benchmark. The authors set several subtasks for graph modification tasks and evaluated several LLMs. Based on this, the authors also propose a novel MAP algorithm.

**Strengths:**

The authors propose a new evaluation of graph modification evaluation.

**Weaknesses:**

The paper is not well organized. What are the input and output of the task? What is the answer format? Does the final answer a graph or some token indicating the added nodes or edges? If the output is the graph itself, how do you build the MAP prompt? Please give some illustrations of your prompt and answer examples.

The experiments lack analysis. For example, why is the remove node dropping while the number of modifications increases while the add node is increasing? Besides, the authors suggest that they imply a CoT prompt, but they have no idea how they do a CoT prompt.

The authors should illustrate how the modified graphs are important for real-world applications or have a connection with other tasks. Otherwise, while lots of LLMs can achieve a very high score, it is not clear what further work we can do on this topic. This part needs further experiments to enhance.

**Questions:**

See weakness

---

> ### Author Response · Authors · 2024-11-24
> **Comment by Authors**
>
> # Organization Issues
>
> We thank the reviewer with pointing out issues with the first paper's organization. As a result, in the new version of the paper, we include Section A.9 that illustrates the form of questions involving zero shot prompting, CoT prompting, and MAP prompting respectively, as well as real model outputs to these prompts. In addition, we include Algorithm 6, which details how the dataset was constructed.
>
> # Lack of Experimental Analysis
>
> We’ve added further analysis to Section A.6 in the Appendix, where we analyze the performance of all models across all final questions and encoding types.
>
> For further analysis, we have also added Section A.7, where we perform an additional ablation study that investigates how varying edge density and graph size impact model performance. Furthermore, we added Section A.8, which details the different errors all benchmarks made on the Add Edge, Remove Edge, Add Node, and Remove Node modifications on the Print Graph task using the Adjacency Matrix encoder (our toughest task).
>
> # Lack of CoT Understanding
>
> Regarding your point on our lack of understanding of CoT prompting, we mentioned previously in Section 4.2.3 that CoT prompting involves “including a list of examples that each demonstrate the reasoning process step-by-step”. Additionally, in Section 5.3.1 we previously mentioned that we provide “examples of detailed reasoning in the prompt”, similarly to how CoT prompting is defined in [1] who first introduced and defined CoT prompting as few-shot prompting that consists of triples: <input, chain of thought, output>. In addition, Figure 37 in Section A.9 in our new paper illustrates how we use CoT prompting.
>
> # Real-world Applications
>
> Thank you for highlighting this key point regarding the relationship between modifying graphs and real-world applications. In the fourth paragraph of the introduction, we previously describe how graph modifications play a central role in dynamic domains such as social networks, where connections are frequently added or removed, and evolving knowledge bases, which must accommodate new or outdated information.
>
> Furthermore, structured data in tables is prevalent across applications like recommendation systems, supply chain management, and relational databases. Graphs are the simplest instantiation of such structured data, with the adjacency matrix effectively functioning as a 2D table of 0s and 1s indicating binary relationships. Therefore, modifying and reasoning on tabular structured data is at least as hard as modifying and reasoning on a graph, making graphs a natural testbed for evaluating how LLMs handle and update structured information.
>
> By focusing on graph modifications, our work provides a benchmark to assess and improve LLMs’ capabilities for tasks that demand precise manipulation and reasoning on evolving data structures, which are critical for real-world applications.
>
> We agree that in this paper, we show that current LLMs have achieved high performance on both the Incident List and Coauthorship encodings, and we would consider graph reasoning problems that use these encodings to be “solved” problems. However, LLMs struggle significantly on the Adjacency Matrix, especially during node removal operations. The Adjacency Matrix encoding requires the LLM to handle dense numerical representations that are both explicit and inherently structured, unlike the more descriptive and naturalistic formats of the Incident List or Coauthorship encodings. In these latter encodings, relationships are expressed through natural language, allowing the model to rely on linguistic patterns and explicit textual cues to infer and modify relationships. By contrast, the adjacency matrix demands that the model infer connections through numerical indexing alone.
>
>
> [1] Wei, Jason, et al. "Chain-of-thought prompting elicits reasoning in large language models." Advances in neural information processing systems 35 (2022): 24824-24837.

---

> > ### Comment · Reviewer_p6sQ · 2024-11-26
> >
> > Thanks for your response. As I mentioned in the original comments: "This part needs further experiments to enhance." I think the experiments may be better. I will keep the score.

---

> ### Author Response · Authors · 2024-11-27
> **Responding to Official Comment**
>
> Dear Reviewer p6sQ,
>
> In your original comment, you mentioned that "illustrat[ing] how the modified graphs are important for real-world applications or have a connection with other tasks...needs further experiments to enhance". Below, we will show how graph modification is important for real-world applications, and why further experiments are not needed to illustrate this relationship. The below explanation has been clarified in the newest version of the paper.
>
> Firstly, we would like to mention that prior work, namely [1] and [2], pose the question of if LLMs can **reason on static graphs**. The authors of these papers define "reasoning" in this sense as the ability of these LLMs to perform **static graph algorithms** (e.g connectivity, shortest path, etc.). A static graph algorithm takes in a static graph and outputs some property of the input graph. Therefore, the authors of [1] and [2] test the "graph reasoning" abilities of LLMs by evaluating if they are able to perform these static graph algorithms.
>
> Please refer to this Simons Institute talk by Monika Henzinger [3] that defines a **“dynamic graph algorithm”**. These algorithms are connected to a myriad of real-world applications, including:
>
> - Social networks: Efficiently updating user connections in platforms like Twitter or Facebook as edges (relationships) and nodes (users) are added or removed.
> - Telecommunication networks: Rapidly reconfiguring connections to maintain communication paths when nodes or links fail.
> - Dynamic routing in transportation: Managing and updating routes in real-time based on traffic conditions, road closures, or new infrastructure.
>
> From the talk and from extensive literature on the topic, a dynamic graph algorithm is composed of two parts:
>
> 1. *“[A] data structure that maintains information about a graph while it is modified through local operations such as edge or vertex insertions or deletions.”*
> 2. The ability to efficiently return properties of the updated graphs without recalculating from scratch after each modification.
>
> In our paper, we pose the question of if LLMs can **reason on dynamic graphs**. In order for an LLM to be able to reason on dynamic graphs, they must be able to do both of the above (accurately maintaining a data structure that undergoes a series of modifications, and efficiently return dynamic properties). As we show in our paper, LLMs struggle greatly on the first point, since they struggle with accurately maintaining one of the most commonly used data structures for representing graphs, the adjacency matrix, as the structure undergoes several key atomic modification operations. Therefore, graph modifications are inextricably linked with the ability to perform dynamic graph algorithms, which are used in a myriad of real world applications, firmly establishing the relationship between graph modifications and real-world applications as well as connections with other tasks.
>
> At this point of the review process, we would appreciate actionable and constructive feedback. **Reviewer p6sQ, could you clarify the following question**:
>
> "Given how we have already "illustrate[d] how the modified graphs are important for real-world applications or have a connection with other tasks" using the definition of dynamic graph algorithms, what exact additional experiments would you like us to perform, and how would these experiments add further clarity on the relationship between graph modification and real-world applications?"
>
> [1] Fatemi, Bahare, Jonathan Halcrow, and Bryan Perozzi. "Talk like a graph: Encoding graphs for large language models." arXiv preprint arXiv:2310.04560 (2023).
>
> [2] Wang, Heng, et al. "Can language models solve graph problems in natural language?." Advances in Neural Information Processing Systems 36 (2024).
>
> [3] https://simons.berkeley.edu/news/dynamic-graph-algorithms-what-we-know-what-we-dont

---

> > ### Comment · Reviewer_p6sQ · 2024-11-27
> >
> > Let’s analyze why I think you need the extra experiments and why I would like to keep my score step by step.
> >
> > 1.	You consider this work to be a benchmark study. Let’s examine what constitutes a benchmark from the perspective of dataset design. In [1], they include 7 datasets spanning 3 domains and evaluate 18 models, including 12 GraphLLM models. In [2], the work incorporates 3 synthetic datasets and 4 real-world datasets, assessing performance across 5 tasks. In contrast, your work exclusively utilizes synthetic datasets with similar task objectives. While you have generated 500,000 questions, they are confined to a narrow domain.
> >
> >
> >
> > 2. If you aim to align your work with studies like NLGraph[4] or Talk Like Graph[3], let’s examine the key insights they derived from their experimental results. In NLGraph, they foud that LLMs have the basic ability on graph reasoning, LLMs are (un)surprisingly brittle to spurious correlations in problem settings. Besides, they also provide a new prompt method called BAG. In Talk like graph, they provide the insights are: Graph encoding functions have significant impact on LLM reasoning; Integer node encoding improves arithmetic performance. These findings were novel at the time of publication. However, what new discoveries does your submission contribute? The observed improvement in overall reasoning ability for LLMs, which enables better performance across different tasks, is already a well-known conclusion.
> >
> >
> >
> > 3. For error analysis, I appreciate that you have addressed some analsysis, but the illustration at this part is still confusion. NLGraph[4] suggests that LLMs struggle with connectivity tasks, particularly in chain graphs. In GraphArena[5], a section is dedicated to analyzing hallucination issues. In your work, have you identified when and why LLMs are likely to fail, providing a deeper understanding of these phenomena? For example, is this related to graph structure? graph structure? or graph density? While you mention several error types in your appendix, their specific meanings, the reasons behind these errors, and their implications are not clearly explained. What are the real-world impacts of these errors, especially if you aim to position your work as having real-world potential? This is why I recommend adding real-world applications and conducting a deeper analysis of the errors to strengthen your claims. If you wish to claim that your work is the first to address dynamic graph modification, please also demonstrate the specific errors LLMs are prone to when working with dynamic graphs.
> >
> > 4. As you metioned, LLMs struggle greatly on the first point, since they struggle with accurately maintaining one of the most commonly used data structures for representing graphs, the adjacency matrix, as the structure undergoes several key atomic modification operations (accurately maintaining a data structure). This is a well-known conclusion from previous work: no LLM has achieved 100% accuracy on any graph-related task. If this conclusion is already established in the literature, why did you choose to focus on graph modification in dynamic graphs? This really wired.
> >
> > Overall, I do not believe this submission is ready due to its limited contribution. To improve the quality of your paper, I suggest including at least one real-world dataset to demonstrate that your synthetic dataset aligns with practical applications. Additionally, you could provide new insights derived from these datasets. Without such additions, I feel this paper offers minimal contribution to the advancement of LLMs in the graph domain.
> >
> >
> > [1] Li Y, Wang P, Zhu X, et al. GLBench: A Comprehensive Benchmark for Graph with Large Language Models[C]//The Thirty-eight Conference on Neural Information Processing Systems Datasets and Benchmarks Track.
> >
> > [2] Zhang Y, Wang H, Feng S, et al. Can LLM Graph Reasoning Generalize beyond Pattern Memorization?[C]//Findings of the Association for Computational Linguistics: EMNLP 2024. 2024: 2289-2305.
> >
> > [3] Fatemi, Bahare, Jonathan Halcrow, and Bryan Perozzi. "Talk like a graph: Encoding graphs for large language models." arXiv preprint arXiv:2310.04560 (2023).
> >
> > [4] Wang, Heng, et al. "Can language models solve graph problems in natural language?." Advances in Neural Information Processing Systems 36 (2024).
> >
> > [5]GraphArena: Benchmarking Large Language Models
> >  on Graph Computational Problems

---

> > > ### Author Response · Authors · 2024-11-29
> > > **Responding to Offical Comment (1/2)**
> > >
> > > Thank you for your response, we greatly appreciate the back-and-forth dialogue and the chance for clarification. We address your points below:
> > >
> > > # 1. Defining a Benchmark Study
> > >
> > > ## Synthetic Data
> > >
> > > It appears that we disagree on “what constitutes a benchmark from the perspective of dataset design.” Both NLGraph [1] and Talk Like a Graph [2] are entirely composed of synthetic data. *The authors of both papers actually point to why they opted to exclusively utilize synthetic data*. In NLGraph, the authors mention that they use synthetic data in order to have more control in varying the complexity of the underlying prompts, and additionally mention that “synthetic graph reasoning problems and their answers are highly unlikely to have exact matches in the training corpora, which makes NLGraph a more robust benchmark towards evaluating language model reasoning.” In Talk Like a Graph, the authors mention that synthetic generation is used to vary properties of the graph structure in order to investigate how these properties “influence the choice of graph encoding function”. Additionally, other work has showcased the potential of synthetic data for evaluating LLMs [3], highlighting that it can be "generated at scale", "tailored to specific requirements", and can "increase the complexity of questions and answers".
> > >
> > > From these papers, it becomes apparent that synthetic data allows us to precisely control the complexity of the underlying problem, avoid train-test contamination, and understand how variations in the underlying data structures we are investigating have downstream effects on performance.
> > >
> > > According to your definition of a benchmark, benchmark datasets must include real-world data, and assess performance on at least 5 different tasks. Therefore, it appears that NLGraph [1] and Talk Like a Graph [2], by your own definition, are **not valid benchmarks**. Would you agree with this claim?
> > >
> > > You also mention the following:
> > >
> > > > I suggest including at least one real-world dataset to demonstrate that your synthetic dataset aligns with practical applications.
> > >
> > > Are you saying that both NLGraph [1] and Talk Like a Graph [2] have **not** demonstrated that their synthetic datasets align with practical applications? If so, then it appears that you believe that prior and seminal work in the field of graph reasoning with LLMs is severely limited. If not, however, then we have already demonstrated that our “synthetic dataset aligns with practical applications”, which clears up this point.
> > >
> > > ## Narrow Task Domain
> > >
> > > You also mention that the tasks found in our dataset have “similar task objectives”, even though we include tasks found in Talk Like a Graph [2] and add an additional task, Print Graph (resulting in 5 different tasks). Therefore, it appears that you are claiming that the tasks found in Talk Like a Graph [2] are “confined to a narrow domain.” Would you agree with this claim?
> > >
> > > # 2. Key Insights from GraphModQA
> > >
> > > Thank you for synthesizing [1] and [2]’s experimental findings. Firstly, you mentioned that “[t]he observed improvement in overall reasoning ability for LLMs, which enables better performance across different tasks, is already a well-known conclusion.” To be clear, we agree with this claim. This claim is not our contribution, and we never claim it to be. It is our motivation for this paper. Please reread the Introduction of the paper.
> > >
> > > In addition, all main “new discoveries” that our submission contributes can be found in the Introduction and Conclusion sections of our paper. Please let us know if you would like us to list them here.
> > >
> > > # 3. Error Analysis
> > >
> > > Section A.5 discusses the relationship between graph structure and error rates. Section A.7 discusses the relationship between graph size, graph density, and error rates. Regarding Section A.8, you mention that for each error type, we do not clearly explain their “specific meanings, the reasons behind these errors, and their implications”. Regarding “specific meanings”, we explicitly define each error type in this section. Therefore, the specific meaning of each error type is clearly established. On “the reasons behind these errors”, we show that the errors’ rates of occurrence are defined on three axis: 1) the LLM used, 2) the current modification type being performed, and 3) the complexity of the problem (defined by the number of modifications to be performed). We plots these relationships in Figures 29-32 in order for the reader to visualize the reasons behind these errors. Finally, on the “implications” of these errors, like we’ve mentioned before, these error rates indicate that LLMs struggle to perform dynamic graph algorithms as they are unable to accurately maintain dynamic data structures, meaning LLMs are unable to perform mainly dynamic graph algorithms that are used across many real-world applications (e.g transportation). In these ways, our appendix clearly “demonstrate[s] the specific errors LLMs are prone to when working with dynamic graphs.”

---

> > > > ### Author Response · Authors · 2024-11-29
> > > > **Responding to Offical Comment (2/2)**
> > > >
> > > > # 4. Lack of Novel Conclusion
> > > >
> > > > We are struggling to understand your last point. Are you saying that the fact that LLMs “struggle with accurately maintaining…the adjacency matrix, as [it] undergoes several key atomic modification operations” is “a well-known conclusion from previous work”? If so, your statement is categorically false, as our paper is the first to evaluate the ability of LLMs to modify adjacency matrices.
> > > >
> > > > Or perhaps, are you saying that because “no LLM has achieved 100% accuracy on any graph-related task”, it is not novel to discover new tasks that LLMs struggle on? If so, it seems that you mainly take issue with the field of graph reasoning with LLMs and the research process itself rather than our paper. For example, NLGraph [1] first establishes graph reasoning problems that LLMs struggle with. Then, Talk Like a Graph [2] builds on [1] by introducing new tasks and encoding functions that models struggle on. Therefore, are you saying that Talk Like a Graph [2] lacks novelty because “no LLM has achieved 100% accuracy on any graph-related task”? Please explain.
> > > >
> > > > [1] Wang, Heng, et al. "Can language models solve graph problems in natural language?." Advances in Neural Information Processing Systems 36 (2024).
> > > >
> > > > [2] Fatemi, Bahare, Jonathan Halcrow, and Bryan Perozzi. "Talk like a graph: Encoding graphs for large language models." arXiv preprint arXiv:2310.04560 (2023).
> > > >
> > > > [3] Liu, Ruibo, et al. "Best practices and lessons learned on synthetic data for language models." arXiv e-prints (2024): arXiv-2404.

---

> > > > > ### Author Response · Authors · 2024-12-01
> > > > > **Invitation for Further Discussion**
> > > > >
> > > > > Dear Reviewer p6sQ,
> > > > >
> > > > > As we come closer to the end of the discussion period, please let us know if you have any thoughts regarding our above comment addressing your concerns. We thank you very much for your efforts thus far.
> > > > >
> > > > > Authors

---

### Official Review · Reviewer_ER9D · 2024-11-02

**Soundness:** 3
**Presentation:** 2
**Contribution:** 3
**Rating:** 3
**Confidence:** 4

**Summary:**

This paper presents an innovative investigation into LLMs' capabilities in graph modification tasks. The authors introduce GraphModQA, a novel benchmark dataset designed to evaluate LLMs' ability to manipulate graphs through operations like adding/removing nodes and edges. Through comprehensive experiments with modern LLMs including GPT-4o-mini, Llama 3.1 405B, Claude 3.5 Sonnet, and o1-mini, they demonstrate that while these models excel at static graph property tasks, they struggle with dynamic graph modifications, particularly when using adjacency matrix representations. The work introduces a new prompting technique called Modify and Print (MAP) prompting, which shows promising improvements in model performance by requiring explicit output of intermediate graph states.

**Strengths:**

This manuscript demonstrates several notable strengths that warrant recognition in the field of LLM research and graph manipulation. The introduction of GraphModQA represents a significant advancement in benchmarking LLM capabilities, particularly through its novel focus on dynamic graph modifications rather than merely static property assessment. The authors have thoughtfully designed their experimental framework, incorporating various graph encoding methods and paying special attention to the previously unexplored adjacency matrix representation. The proposed Modify and Print (MAP) prompting technique shows genuine innovation in addressing the challenges of graph modification tasks, with empirical results demonstrating measurable improvements across different model architectures. Furthermore, the comprehensive evaluation across multiple state-of-the-art models (GPT-4o-mini, Llama 3.1 405B, Claude 3.5 Sonnet, and o1-mini) provides valuable insights into the relative capabilities of different architectures in handling structured data modifications. The authors' systematic approach to testing different modification types (Add Edge, Remove Edge, Add Node, Remove Node, and Mix) offers a thorough exploration of model behaviors under various graph transformation scenarios, while their detailed analysis of performance degradation patterns provides actionable insights for future research directions in improving LLM capabilities for graph manipulation tasks.

**Weaknesses:**

1) The authors claim that modern LLMs "excel at basic graph primitives" but don't provide sufficient empirical evidence comparing performance across model generations.
2) The motivation for choosing graph modification as a benchmark task lacks rigorous justification. While intuitively sensible, there's no formal analysis of why this specifically tests model capabilities better than existing benchmarks.
3) The modification sequence function m(M,k) needs stronger theoretical grounding. The selection of modification types appears arbitrary without formal justification of their completeness or representativeness.
4) The analysis of prompt consistency lacks statistical validation. The authors need to include confidence intervals and significance tests for their performance claims.
5) The experimental setup for comparing different prompting strategies has several problems: No ablation studies to isolate the impact of different components; Missing justification for the choice of 250 graphs; Lack of error analysis on failure cases.
6) The hyperparameter selection process for MAP prompting isn't documented. Critical details about implementation choices are missing.
7) The model selection criteria need better documentation, particularly regarding architecture choices.
8) its presentation suffers from severe readability issues that significantly undermine its potential impact. The visual presentation is particularly problematic - all figures suffer from inadequate formatting, with Figure 2's experimental results being especially difficult to interpret due to its cluttered layout, minuscule font sizes, and insufficient caption detail.

**Questions:**

1) Can you provide statistical significance tests for the performance improvements claimed with MAP prompting?
2) For Section 5.2: How does the choice of 250 test graphs impact result reliability? What power analysis was performed?
3) What ablation studies were conducted to verify the individual components' contributions to performance?
4) For Table 1: Why were these specific baselines chosen? How do they compare to other recent graph-focused LLM evaluations?

---

> ### Author Response · Authors · 2024-11-24
> **Comment by Authors**
>
> # Lack of Evidence on Graph Primitive Performance
>
> Regarding your point on the lack of empirical evidence on the performance across model generations on basic graph primitives, Table 1 in [1], the paper that we cite as inspiring our work, shows the weak performance on older models (particularly those from the PaLM family) on basic graph primitives (e.g Node Count, Node Degree, etc.). In addition, in Tables 1 and 2 in the Appendix of our paper, we show that current SOTA methods clearly outperform older models on basic graph primitives. In Table 2, we evaluate Palm 2-L on the adjacency matrix encoding, and show that SOTA LLMs strongly outperform Palm 2-L on basic graph primitives on this encoding.
>
> # Motivation of GraphModQA
>
> In the introduction, particularly in the fourth paragraph, we justify GraphModQA as a benchmark by highlighting its unique requirements compared to GraphQA. GraphModQA tasks involve iterative graph modifications, such as adding or removing nodes or edges, followed by reasoning about the resulting graph structure. Successfully solving these tasks demands precise state tracking and manipulation of an evolving internal graph representation. Unlike GraphQA, where models reason over static graphs with fixed structures, GraphModQA introduces a compounding challenge: each modification dynamically alters the graph, requiring the model to not only update its representation accurately but also maintain consistency across a sequence of interdependent operations, creating a more complex reasoning landscape. Furthermore, GraphModQA combines the inherent difficulty of dynamic state maintenance with high-level reasoning about the final modified graph, making it significantly more rigorous in evaluating a model's capability to handle evolving graph structures. We have made additions to this paragraph in our new version of the paper in order to make these points clearer.
>
> # Arbitrariness of Modification Types
>
> Regarding your point on the justification of the modification types, please firstly refer to the following paper [2]. Here is an excerpt from this paper that justifies our modification types:
>
>  *“(static) graphs and networks involve the following entities: V (a set of nodes), E (a set of edges), f (map vertices to numbers), and g (map edges to numbers). A dynamic graph is obtained when any of these four entities changes over time. Thus, there are four basic kinds of dynamic graphs.*
>
> - *In a node-dynamic graph or digraph, the set V varies with time. Thus, some nodes may be added or removed. When nodes are removed, the edges (or arcs) incident with them are also eliminated.*
>
> - *In an edge-dynamic (or arc-dynamic) graph or digraph, the set E varies with time. Thus, edges may be added or deleted from the graph or digraph.*
>
> - *In a node weighted dynamic graph, the function f varies with time; thus, the weights on the nodes also change.*
>
> - *In an edge or arc weighted dynamic graph or digraph, the function g varies with time.*
>
> - *In fully weighted dynamic graph, both functions f and g may vary with time.*
>
> *All combinations of the above types can occur.”*
>
> Therefore, in dynamic graphs, “nodes may be added or removed”, and “edges may be added or deleted”, which is why we chose these atomic graph modification operations.
>
> In addition, please refer to this Simons Institute talk by Monika Henzinger [3] that defines a “*dynamic graph algorithm*” as “*a data structure that maintains information about a graph while it is modified through local operations such as edge or vertex insertions or deletions.*” Therefore, in order to modify a graph, one can either add edges, remove edges, add nodes, or remove nodes, which again are all modification operations that we analyze in our paper.
>
> # Choice of 250 Graphs
>
> We first chose 250 graphs to get a comprehensive dataset while keeping monetary costs in mind, as 250 graphs created over 400k examples (as explained in Section 4.3), which lead to over 1.6 million independent evaluations across all four benchmark models. Secondly, through statistical analysis and the statistical significance table we add below, we were able to show statistical significance of performance improvements with MAP prompting using 250 graphs.
>
> # Lack of Ablation Studies
>
> In Section A.7, we include an ablation study where we investigate the effect of edge density and the number of nodes in the graph on performance, finding that both variables can significantly alter performance depending on both the modification type and the number of modifications to be performed. We will evaluate the other LLMs in the final version of the paper.
>
> [1] Fatemi, Bahare, Jonathan Halcrow, and Bryan Perozzi. "Talk like a graph: Encoding graphs for large language models." arXiv preprint arXiv:2310.04560 (2023).
>
> [2] Harary, Frank, and Gopal Gupta. "Dynamic graph models." Mathematical and Computer Modelling 25.7 (1997): 79-87.
>
> [3] https://simons.berkeley.edu/news/dynamic-graph-algorithms-what-we-know-what-we-dont

---

> > ### Author Response · Authors · 2024-11-24
> > **Continuation of First Comment by Authors**
> >
> > # Lack of Statistical Validation
> >
> > We thank the reviewer on recognizing the importance of statistical validation. In the below table, we attach statistical significance tests for the performance improvements claimed by MAP prompting. In the paper, we mentioned that MAP prompting performed particularly well on edge-related tasks. This can be seen from the table, where all tests on edge-related task confirm statistical significance with the exception of Remove Edge involving GPT 4o-mini. We additionally observe statistically significant improvements on the Remove Node modification, the most challenging modification, for both Claude 3.5 Sonnet and Llama 3.1 405B. These tests help support the performance improvements claims made in the paper regarding MAP prompting.
> > | Model               | Modification Task | P-value     | T-statistic       | Confidence Intervals     | Cohen's d | Effect Size      | Statistical Significance   |
> > |---------------------|-------------------|-------------|-------------------|--------------------------|-----------|------------------|---------------------------|
> > | Claude 3.5 Sonnet  | add_edge          | 1.4838e-03  | 7.7633           | (8.1316, 16.5084)       | 3.6463    | Large            | Yes                       |
> > | Claude 3.5 Sonnet  | remove_edge       | 2.5514e-02  | 3.4730           | (1.2702, 8.4898)        | 1.6758    | Large            | Yes                       |
> > | Claude 3.5 Sonnet  | add_node          | 5.2913e-01  | -0.6882          | (-0.8865, 0.4065)       | -0.4602   | Small            | No                        |
> > | Claude 3.5 Sonnet  | remove_node       | 2.7399e-02  | 3.3952           | (-10.7770, 44.6970)     | 0.7580    | Medium           | Yes                       |
> > | Claude 3.5 Sonnet  | mix               | 7.0888e-03  | 5.0784           | (5.4020, 41.1580)       | 1.6142    | Large            | Yes                       |
> > | Llama 3.1 405B     | add_edge          | 9.7713e-03  | 4.6349           | (-7.4607, 22.1807)      | 0.6156    | Medium           | Yes                       |
> > | Llama 3.1 405B     | remove_edge       | 1.1109e-02  | 4.4660           | (-9.6348, 19.7148)      | 0.4257    | Small            | Yes                       |
> > | Llama 3.1 405B     | add_node          | 5.7339e-01  | 0.6124           | (-0.5361, 1.0161)       | 0.3833    | Small            | No                        |
> > | Llama 3.1 405B     | remove_node       | 6.7974e-03  | 5.1387           | (-7.1415, 17.5415)      | 0.5223    | Medium           | Yes                       |
> > | Llama 3.1 405B     | mix               | 7.2680e-02  | 2.4211           | (-22.5382, 29.8982)     | 0.1740    | Negligible       | No                        |
> > | o1-mini            | add_edge          | 5.5681e-03  | 5.4332           | (2.3848, 18.4152)       | 1.6084    | Large            | Yes                       |
> > | o1-mini            | remove_edge       | 4.6479e-02  | 2.8482           | (-0.7348, 15.9348)      | 1.1303    | Large            | Yes                       |
> > | o1-mini            | add_node          | 2.1289e-01  | -1.4803          | (-19.2572, 4.3092)      | -0.7863   | Medium           | No                        |
> > | o1-mini            | remove_node       | 4.1586e-02  | -2.9592          | (-25.3689, 3.3129)      | -0.9532   | Large            | Yes                       |
> > | o1-mini            | mix               | 7.6217e-01  | -0.3240          | (-17.4019, 15.7219)     | -0.0629   | Negligible       | No                        |
> > | GPT-4o mini        | add_edge          | 1.1272e-02  | 4.4471           | (-15.0958, 22.7758)     | 0.2514    | Small            | Yes                       |
> > | GPT-4o mini        | remove_edge       | 3.1840e-01  | 1.1387           | (-21.1317, 29.2917)     | 0.2006    | Small            | No                        |
> > | GPT-4o mini        | add_node          | 4.0834e-01  | -0.9228          | (-38.8715, 17.1035)     | -0.4821   | Small            | No                        |
> > | GPT-4o mini        | remove_node       | 2.8659e-01  | 1.2285           | (-3.1552, 5.2552)       | 0.3095    | Small            | No                        |
> > | GPT-4o mini        | mix               | 2.5678e-01  | 1.3217           | (-21.9611, 26.0571)     | 0.1057    | Negligible       | No                        |
> >
> > # MAP Hyperparameter Selection
> >
> > In the new version of our paper, we have added Figure 35 illustrating MAP prompting, which involves simply appending “For each operation, write out the entire resulting graph.” to the end of the prompt. As a result, there are no hyperparameters involved for MAP prompting.

---

> ### Author Response · Authors · 2024-11-24
> **Further Continuation of First Comment**
>
> # Model Selection Criteria
>
> We selected GPT-4o-mini, Llama 3.1 405B, Claude 3.5 Sonnet, and o1-mini because they represent the latest advancements in LLMs, covering a spectrum of architectures and model sizes. All four LLMs achieve high performance across many benchmarks, including LiveBench [4].
>
> # Presentation Issues
>
> Thank you for pointing out these issues with our presentation. In the new version of our paper, all figures have increased readability.
>
> # Lack of Error Analysis
>
> We thank the reviewer for suggesting this addition. We added Section A.8, which details the different errors all benchmarks made on the Add Edge, Remove Edge, Add Node, and Remove Node modifications on the Print Graph task using the Adjacency Matrix encoder (our toughest task). On edge-related modifications, we find that models make interesting hallucinations when modifying indices, as they tend to incorrectly modify adjacent indices as well that weren’t specified in the prompt. On Add Node, while Claude 3.5 Sonnet and Llama 3.1 405B tend to perform well, o1-mini and GPT 4o-mini often return objects that are not mathematically valid matrices, and in our analysis these objects mostly take the form of a list of rows with inconsistent column counts. On Remove Node, all models struggle significantly, and return a matrix whose internal connections deviate substantially from the solution matrix.
>
> # Question 4
>
> These baselines are from [1], and as such, they are among the most recent graph-focused LLM evaluations. As we stated in the Related Works section, these baselines benefit from their straightforward and fundamental nature compared to other graph-focused LLM evaluations.
>
> [4] White, Colin, et al. "Livebench: A challenging, contamination-free llm benchmark." arXiv preprint arXiv:2406.19314 (2024).

---

> > ### Author Response · Authors · 2024-11-27
> > **Coming close to end of PDF Revision**
> >
> > Dear Reviewer ER9D,
> >
> > We thank you for your feedback thus far. As we are approaching the PDF Revision deadline, please let us know if you have any comments or concerns that we can directly integrate into the paper.
> >
> > Thank you,
> > Authors

---

> > > ### Comment · Reviewer_ER9D · 2024-11-28
> > >
> > > Thank you for response. After careful review, I still think there are several issues that need to be addressed.
> > >
> > > > "Given these advances, we propose a more challenging evaluation problem: graph modification, a foundational, interpretable, and non-trivial problem in which an LLM must determine the outcome of adding or deleting a given sequence of nodes or edges..."
> > >
> > > This foundational claim lacks rigorous justification. The paper does not demonstrate why graph modification tasks specifically test LLM capabilities better than existing benchmarks.
> > >
> > > > "We define m(M,k) as the modification sequence function, which outputs a sequence of k modifications of type M to be performed on G."
> > >
> > > The selection of modification types appears arbitrary without formal proof of completeness or representativeness of real-world graph modifications.
> > >
> > > > "The difficulty with the Remove Node modification can likely be attributed to the challenges associated with managing the adjacency matrix representation"
> > >
> > > The error analysis remains largely descriptive without systematic categorization or quantitative backing.
> > >
> > > Your paper would significantly benefit from a more rigorous theoretical grounding explaining why graph modifications specifically serve as better indicators of LLM capabilities compared to existing benchmarks. While you mention Harary and Gupta's work on dynamic graph models, you haven't established why this particular set of operations provides unique insights into LLM reasoning. The ablation studies currently focus only on edge density and graph size, missing investigations into how different model architectures handle various graph representations, especially for the adjacency matrix encoding which you claim is particularly challenging. Though you've added statistical tests in your response, the main paper lacks a comprehensive statistical framework to validate the claimed improvements from MAP prompting, particularly regarding the significance of performance differences across modification types.
> > >
> > > The paper's presentation remains a barrier to understanding your contributions, and many figures are still problematic with overcrowded or over-spaced plots and insufficient captions, making it difficult to interpret the relative performance of different models. While you've included an error analysis section, it primarily describes observed behaviors without providing a systematic categorization framework that could guide future improvements in LLM architectures for graph manipulation tasks. I will keep my score.

---

> > > > ### Author Response · Authors · 2024-11-29
> > > > **Response to Official Comment**
> > > >
> > > > # Foundational Claim Lacks Rigorous Justification
> > > >
> > > > > This foundational claim lacks rigorous justification. The paper does not demonstrate why graph modification tasks specifically test LLM capabilities better than existing benchmarks.
> > > >
> > > > As you made this point nearly verbatim in your original review, we have already responded to this point in our original rebuttal. Please refer to the heading “Motivation of GraphModQA”.
> > > >
> > > > # Arbitrariness of Modification Types
> > > >
> > > > > The selection of modification types appears arbitrary without formal proof of completeness or representativeness of real-world graph modifications.
> > > >
> > > > Again, as you made this point nearly verbatim in your original review, we have also responded to this point above. Please refer to the heading “Arbitrariness of Modification Types”.
> > > >
> > > > # Error Analysis
> > > >
> > > > > The error analysis remains largely descriptive without systematic categorization or quantitative backing.
> > > >
> > > > > While you've included an error analysis section, it primarily describes observed behaviors without providing a systematic categorization framework that could guide future improvements in LLM architectures for graph manipulation tasks.
> > > >
> > > > You claim that the error analysis does not have 1) “systematic categorization”, and 2) “quantitative backing”.
> > > >
> > > > 1. “Systematic categorization”: In Sections A.8.1 to A.8.4, we name and categorize each of the different types of errors on the Add Edge, Remove Edge, Add Node, and Remove Node modifications respectively. We also define these errors and compare the rates at which different models make these errors.
> > > > 2. “Quantitative backing”: Figures 29-32 plot the number of times each of the four LLMs make these errors as a function of the number of modification steps. These plots quantitively support the statements made in this section.
> > > >
> > > > # Ablation Studies
> > > >
> > > > > The ablation studies currently focus only on edge density and graph size, missing investigations into how different model architectures handle various graph representations, especially for the adjacency matrix encoding which you claim is particularly challenging.
> > > >
> > > > Please clarify how you define “graph representations”. If you are referring to different graph types/structures (i.e outside of ER graphs, such as Star/Path graphs), please refer to Section A.5. If you are referring to different graph encodings (e.g Incident, Coauthorship), please refer to Section A.6.
> > > >
> > > > # Statistical Validation
> > > >
> > > > > Though you've added statistical tests in your response, the main paper lacks a comprehensive statistical framework to validate the claimed improvements from MAP prompting, particularly regarding the significance of performance differences across modification types.
> > > >
> > > > Please clarify this point. Are you saying that even though we have performed a statistical analysis that explicitly analyzes “the significance of performance differences across modification types”, and that this analysis can easily be added into the future version of the paper, you are still marking us down for not having this analysis in our second version of the paper?
> > > >
> > > > # Presentation issues
> > > >
> > > > > The paper's presentation remains a barrier to understanding your contributions, and many figures are still problematic with overcrowded or over-spaced plots and insufficient captions, making it difficult to interpret the relative performance of different models.
> > > >
> > > > It would be helpful for us and this discussion if you could list the particular figures that remain "problematic" in the paper.

---

> > > > > ### Author Response · Authors · 2024-12-01
> > > > > **Invitation for Further Discussion**
> > > > >
> > > > > Dear Reviewer ER9D,
> > > > >
> > > > > As we come closer to the end of the discussion period, please let us know if you have any thoughts regarding our above comment addressing your concerns. We thank you very much for your efforts thus far.
> > > > >
> > > > > Authors

---

### Official Review · Reviewer_KpyE · 2024-11-04

**Soundness:** 2
**Presentation:** 2
**Contribution:** 2
**Rating:** 3
**Confidence:** 5

**Summary:**

The paper explores the capability of LLMs in handling graph modification tasks, which involve adding or deleting nodes or edges and computing properties on the modified graph. It introduces GraphModQA, a benchmark dataset with question-answer pairs aimed at testing LLMs’ skills in graph manipulation and dynamic reasoning. The authors find that while current state-of-the-art LLMs handle static graph tasks well, their performance drops significantly with graph modifications, especially when using adjacency matrix representations. To address this, the authors propose Modify and Print prompting, a method where models output intermediate adjacency matrices, leading to improved performance.

**Strengths:**

- Clear write-up, which is easy to follow and get the core idea for readers.
- The proposed GraphModQA may serve as one of the important benchmarks for LLMs solving graph problems.

**Weaknesses:**

- Presentation. The experiment part in the main papers lacks clear tables to show how different LLMs perform on the GraphModQA benchmark. The authors put a lot of large figures here, but they are less informative and are consuming too much of the space. Especially, the presentation makes the experiment part very difficult to follow and grasp the key findings.
- Lack of detailed presentation and showcasing of the proposed benchmark. After reading the paper, I still have no idea how many questions are there in GraphModQA in total, and what the forms of these questions are.
- The motivation of leveraging LLMs for graph modification remains unclear and confusing. Generally the task is trivial and you can swiftly and smoothly add some nodes or edges to the graph, by simply accessing to tools like PyG or NetworkX. Even if we would like LLMs to acquire such skill, it is more reasonable that if we directly add such feature in a tool-use style, especially for a trivial task, which can be done by a single function call.
- The proposed MAP method centers around CoT prompting for the certain task, and is considered less novel.

**Questions:**

Please see weaknesses.

---

> ### Author Response · Authors · 2024-11-24
> **Comment by Authors**
>
> # Presentation
> We appreciate the reviewer's point regarding the presentation of the paper. Firstly, we have uploaded a new paper that includes clearer figures. Regarding your point on the lack of clear tables to compare the performance across different LLMs on GraphModQA, we don’t include these tables because we want to explicitly show how performance changes as a function of modification length. For the Print Graph task, Figures 2 show the performance of o1-mini, Claude 3.5 Sonnet, Llama 3.1 405B, and GPT-4o-mini respectively on the adjacency matrix encoding, and Figures 11-24 in Section A.6 of the Appendix compare the performance of all four benchmark LLMs across other final questions and graph encodings. Additionally, Section A.6 includes further analysis on the results of these experiments.
>
> # Lack of Showcasing of GraphModQA
> We appreciate the feedback regarding the presentation of the GraphModQA benchmark. The new version of the paper provides a more comprehensive illustration of the benchmark. Specifically, we have added Section 4.3, which details the total number of questions in GraphModQA (468,750 ), and how this number is derived.
>
> # Examples of Questions:
> Regarding the forms of individual questions, we have added Section A.9 which includes Figures 33 to 38. These figures illustrate examples of input prompts and model outputs corresponding to zero-shot prompting, CoT prompting, and MAP prompting.
>
> # Unclear motivation
> Thank you for highlighting this. The first important point to make is that you’re of course correct in that tools like PyG and NetworkX provide efficient and reliable implementations for many graph tasks, let alone modification tasks, in a single function call. However, there is already a growing literature that looks at LLMs' intrinsic reasoning abilities on graphs (e.g., [1] and [2]), and as the literature explains, they are not advocating for replacing these established tools with large language models (LLMs) for practical graph tasks. Instead, they aim to evaluate LLMs’ reasoning capabilities within the context of graph tasks as a way to better understand and quantify their strengths and limitations in multi-step reasoning scenarios.
>
> We recognize a critical gap in the existing literature, where prior work has largely focused on basic static graph properties. Therefore, we add to this literature in a qualitatively new way by introducing the concept of graph modification as a more dynamic, multi-step reasoning challenge. This approach aligns with the broader goal of assessing and advancing LLMs’ ability to handle evolving, structured data representations, which is critical for domains like social network analysis and real-time decision-making in dynamic systems.
>
> # Lack of novelty of MAP prompting
> Regarding your point on the similarity between MAP prompting and CoT prompting, there have been several different iterations of CoT. [3] first introduced CoT prompting by defining it as few-shot prompting consisting of triples: <input, chain of thought, output>. Our proposed MAP prompting, on the other hand, simply appends the phrase “For each operation, write out the entire resulting graph.” to the prompt. Therefore, MAP prompting is distinct and purpose-built for dynamic graph reasoning tasks, and does not center around CoT prompting.
>
> Another iteration of CoT, Zero-shot-CoT [4], simply appends the phrase “Let’s think step by step.” to the prompt. While similar to MAP prompting in the sense that both encourage intermediate reasoning steps and involve appending simple phrases, MAP prompting differs from Zero-shot-CoT by requiring the explicit generation and output of the intermediate graph at each step of the reasoning process. This explicit instruction focusing on graph state tracking ensures that the model maintains a consistent internal representation of the evolving graph, which is particularly critical for tasks involving multiple dynamic modifications. Therefore, MAP prompting introduces a novel approach tailored to dynamic graph reasoning.
>
> [1] Fatemi, Bahare, Jonathan Halcrow, and Bryan Perozzi. "Talk like a graph: Encoding graphs for large language models." arXiv preprint arXiv:2310.04560 (2023).
>
> [2] Wang, Heng, et al. "Can language models solve graph problems in natural language?." Advances in Neural Information Processing Systems 36 (2024).
>
> [3] Wei, Jason, et al. "Chain-of-thought prompting elicits reasoning in large language models." Advances in neural information processing systems 35 (2022): 24824-24837.
>
> [4] Kojima, Takeshi, et al. "Large language models are zero-shot reasoners." Advances in neural information processing systems 35 (2022): 22199-22213.

---

> > ### Comment · Reviewer_KpyE · 2024-11-25
> >
> > Thanks for the authors' rebuttal. I agree with the idea that *"they are not advocating for replacing these established tools with large language models (LLMs) for practical graph tasks. Instead, they aim to evaluate LLMs’ reasoning capabilities within the context of graph tasks as a way to better understand and quantify their strengths and limitations in multi-step reasoning scenarios."*.
> >
> > I would feel more convinced if the paper is presented in a benchmarking version, especially on how LLMs can solve such graph problems by multi-step reasoning and reflection. However, in that way, we would need additional efforts like justifications on why such a strategy would accurately reflect the capacities, or comparing GraphModQA with other reasoning benchmarks.
> >
> > In all, I feel the reasoning examining perspective is more reasonable than how the paper is positioned now. I would keep my vote for the current version.

---

> > > ### Author Response · Authors · 2024-11-26
> > > **Replying to Official Comment by Reviewer KpyE**
> > >
> > > We thank the reviewer for their timely response. To the best of our knowledge, our paper is already presented as a benchmark, where we evaluate the multi-step reasoning abilities of LLMs by using various prompting methods: zero-shot, CoT (with a variable number of examples), and our novel MAP prompting method (which requires the model to explicitly reason on intermediate graphs). Therefore, we are already taking a “reasoning examining perspective”. In addition, in our paper we do compare GraphModQA to other reasoning benchmarks in the Related Works section, in particular GraphQA [1], a benchmark that evaluates the reasoning abilities of LLMs on static graph problems, as opposed to GraphModQA which evaluates the multi-step, dynamic reasoning abilities of LLMs.
> > >
> > > We believe that our paper already tackles these points you mentioned in your response. Therefore, we would like to ask the reviewer the following clarifying questions:
> > >
> > > 1. You mention that you’d feel more convinced “if the paper is presented in a benchmarking version”. However, in your summary, you refer to GraphModQA as a “benchmark dataset with question-answer pairs aimed at testing LLMs’ skills in graph manipulation and dynamic reasoning.” Given that we compare performances on GraphModQA, in what way is our paper not “presented in a benchmarking version”?
> > > 2. In our paper, we explicitly compare the ability of four SOTA LLMs to solve graph modification problems by comparing their performance on GraphModQA, and by performing a detailed error analysis on these LLMs in Section A.8. Therefore, as you mentioned in your response that you would prefer if the paper examined “how LLMs can solve such graph problems by multi-step reasoning and reflection”, could you elaborate on how our approach of evaluating the (in-context learning) abilities of these SOTA LLMs on GraphModQA, a dataset which includes problems that are composed of multiple reasoning steps, fails to achieve this?
> > > 3. As we have already compared GraphModQA to other (graph) reasoning benchmarks in our Related Works section, what other reasoning benchmarks are we not comparing to?
> > >
> > >
> > > [1] Fatemi, Bahare, Jonathan Halcrow, and Bryan Perozzi. "Talk like a graph: Encoding graphs for large language models." arXiv preprint arXiv:2310.04560 (2023).

---

> > > > ### Author Response · Authors · 2024-11-29
> > > > **Invitation for Further Discussion from Reviewer KpyE**
> > > >
> > > > Dear Reviewer KpyE,
> > > >
> > > > As we come closer to the end of the discussion period, please let us know if you have any thoughts regarding our above comment addressing your concerns. We thank you very much for your efforts thus far.
> > > >
> > > > Authors

---

> > > > > ### Author Response · Authors · 2024-12-01
> > > > > **Invitation for Further Discussion**
> > > > >
> > > > > Dear Reviewer KpyE,
> > > > >
> > > > > As we are drawing very close to the end of the discussion period, please let us know if you have any further comments about the paper.
> > > > >
> > > > > Authors

---

### Meta-Review · Area_Chair_9Vjr · 2024-12-20

**Metareview:**

This paper introduces GraphModQA, a benchmark for evaluating LLMs’ graph modification capabilities, and proposes a Modify and Print (MAP) prompting method. All the reviewers gave reject recommendations. The main concerns include:

1. Weak Motivation: The connection between graph modification tasks and real-world applications is insufficiently justified. The paper does not convincingly demonstrate why these tasks are essential for evaluating LLM reasoning capabilities.

2. Limited Contribution: The proposed benchmark and MAP prompting are incremental, offering minimal novelty compared to existing methods and datasets. The tasks are straightforward and do not meaningfully challenge LLMs beyond current benchmarks.

3. Dataset and Analysis Gaps: The benchmark is entirely synthetic, lacking diversity or alignment with real-world graph problems. The error analysis is superficial, and critical dataset details and insights are missing or poorly presented.

**Additional Comments On Reviewer Discussion:**

The rebuttal addressed some concerns but failed to resolve foundational issues. To improve, the work should incorporate real-world datasets, provide deeper insights, and enhance its presentation. As it stands, the paper offers limited advancement for LLMs in graph reasoning and is not ready for acceptance.

---

### Decision · Program_Chairs · 2025-01-22

Reject